



# H2CM (v1.0): hybrid modeling of global water–carbon cycles constrained by atmospheric and land observations

Zavud Baghirov[1,2], Markus Reichstein[1,5], Basil Kraft[4], Bernhard Ahrens[1], Marco Körner[2,3], and Martin Jung[1]

[1]Department of Biogeochemical Integration, Max Planck Institute for Biogeochemistry, Jena, Germany
[2]Department of Aerospace and Geodesy, TUM School of Engineering and Design, Technical University of Munich (TUM), Germany
[3]Munich Data Science Institute, Technical University of Munich (TUM), Munich, Germany
[4]ETH Zurich, Environmental Systems Science, Zurich, Switzerland
[5]ELLIS Unit Jena at Michael-Stifel-Center Jena for Data-driven and Simulation Science, Jena, Germany

**Correspondence:** Zavud Baghirov (zbaghirov@bgc-jena.mpg.de)

**Abstract.** We present the Hybrid Hydrological Carbon Cycle Model (H2CM)—a global model that couples the terrestrial water and carbon cycles by integrating a process-informed deep learning approach with observational constraints for the water and carbon cycles. H2CM extends the hybrid hydrological model with vegetation (H2MV) to represent key terrestrial carbon fluxes, including gross primary productivity (GPP), autotrophic and heterotrophic respiration at daily resolution and
1-degree spatial scale. H2CM uses neural networks to learn and predict ecosystem properties governing water and carbon fluxes, such as carbon and water use efficiencies and basal respiration rate. H2CM uniquely combines multiple observational constraints synergistically: on top of hydrological and vegetation data constraints on terrestrial water storage variations, snow water equivalent, evapotranspiration, runoff and fraction of photosynthetically active radiation, the carbon cycle is informed by an observation-based GPP product, and net ecosystem exchange (NEE) from satellite and in-situ based atmospheric $CO_2$
inversion datasets. H2CM reproduces the seasonal and interannual dynamics of carbon fluxes well. H2CM outperforms both purely data-driven models as well as state-of-the-art process-based model ensembles in capturing NEE seasonality, especially in challenging regions such as the South American tropics and Southern Africa. Moreover, H2CM reveals emergent spatial patterns in precipitation use efficiency, light use efficiency, and water-carbon coupling, consistent with empirical ecological understanding. Notably, we show that H2CM learns to represent the rain pulse effect on respiration in dry regions, which is
often not well reproduced by global models. H2CM represents a key step toward a new generation of hybrid land surface models, with planned extensions to include the energy cycle.

## 1 Introduction

The water-carbon cycle is a critical component of Earth's ecosystems, significantly influencing our understanding of climate, water resources, and carbon dynamics. Previous studies have demonstrated that global water and carbon cycles are strongly
interconnected (Jung et al., 2017; Humphrey et al., 2018).





Global water and carbon cycle processes are typically modeled using two main strategies: data-driven modeling and process-based modeling (PBM). Data-driven approaches often involve machine learning (ML) to estimate quantities related to the water or carbon cycle (Dou et al., 2018; Shi et al., 2024; Tian et al., 2023; Jung et al., 2019, 2020; Nelson et al., 2024). ML can learn from observational data with minimal prior knowledge, relaxing uncertain assumptions and potentially leading to new insights

and accurate predictions. This becomes increasingly relevant as the volume of Earth observation data grows (Huntingford et al., 2019; Rolnick et al., 2022; Schneider et al., 2017; Eyring et al., 2024). However, a significant caveat of using ML to explain the Earth system is that these models are very difficult to interpret in terms of learned intermediate processes and mechanisms (e.g., they function as a "black box"). Additionally, they may suffer from extrapolation issues and they do not guarantee adherence to well-established process-knowledge (Shen et al., 2023; Reichstein et al., 2019).

Another important approach to modeling water-carbon cycle processes is the use of PBMs (Le Quéré et al., 2012; Sitch et al., 2015, 2024). Unlike the ML strategy, PBMs represent process-understanding explicitly and adhere to fundamental laws, such as mass conservation. By design, PBMs' simulations can output various diagnostic variables that are easy to interpret and help to understand drivers of water-carbon cycle variations. However, PBMs abstract the complex processes governing water and carbon cycles and require numerous assumptions about processes due to incomplete process knowledge. Thus, assumptions

and modeling approaches vary across different PBMs, while this uncertainty is reflected in significant inter-model spread of simulations (O'Sullivan et al., 2022). Additionally, unlike ML approaches, PBMs are not designed for fully exploiting the growing volume of Earth observations (Nearing et al., 2021; Shen et al., 2018; Kraft et al., 2021).

A novel approach to modeling global water and carbon cycles—hybrid modeling—has recently emerged. This approach combines ML and PBM within a single framework and aims for leveraging the advantages of both while mitigating their

challenges. For instance, hybrid modeling can replace uncertain parameters or process representations of a PBM with ML estimations, while retaining established process knowledge (e.g., mass balance) in the PBM formulations. This effectively reduces the physical inconsistencies of ML, as ML predictions must pass through process formulations that constrain them to obey process knowledge and maintain physical units. Hybrid modeling also reduces the need for prior assumptions about PBM's uncertain components, if they can be learned from data. Consequently, hybrid modeling allows for leveraging the

growing volume of Earth observations through its ML component while maintaining physical plausibility through its PBM component (Eyring et al., 2024; Reichstein et al., 2019; Shen et al., 2023). However, it is important to note that this is still a young and evolving field and most current works on hybrid modelling are demonstration-style.

A major coupling mechanism between the global water and carbon cycles is related to the carbon-water trade-off during photosynthesis. As plants open their stomata to fix carbon dioxide from the atmosphere they are transpiring water back into the

atmosphere (Katul et al., 2012; Heimann and Reichstein, 2008). Soil moisture is another nexus of the water and carbon cycles, as it regulates both photosynthesis (Humphrey et al., 2018) and heterotrophic respiration (Zhang et al., 2018). However, many aspects and details of the coupled water-carbon cycles remain highly uncertain in PBMs.

In this study, we introduce the hybrid hydrological-carbon cycle model (H2CM), building on the hybrid hydrological model with vegetation (H2MV, Baghirov et al. (2025)). H2MV combines a hydrological model with deep learning (DL) (LeCun et al.,

2015), while being constrained by an array of complementary observational data streams on terrestrial water storage (TWS)





variations, snow water equivalent (SWE), evapotranspiration (ET), runoff and fraction of photosynthetically active radiation (fAPAR). H2CM extends this approach to include the carbon cycle, where transpiration and gross primary productivity (GPP) are explicitly coupled through neural network (NN) predictions of water use efficiency (WUE). H2CM also represents carbon use efficiency (CUE) and basal respiration rate (Rb) by NNs to simulate respiration processes. To improve plausibility and causality of NN predictions we couple separate neural networks, which receive individual sets of meaningful inputs for their task, rather than using one large network that receives all inputs to make all estimations. The model simulates spatio-temporal dynamics of GPP and net primary productivity (NPP), heterotrophic respiration (Rh), and thus net ecosystem exchange (NEE) representing the net carbon exchange between land and atmosphere. H2CM's GPP estimations are directly constrained using observation-based data, while NEE estimations are constrained using atmospheric inversion products.

This study has the the overall objectives to describe and evaluate H2CM. To this end we 1) evaluate the model's performance against withhold data constraints; 2) compare the model's GPP, TER, and NEE simulations with state-of-the-art data-driven and process-based models; 3) assess the plausibility of the model's learned global patterns for precipitation, water, carbon, and light use efficiencies; 4) assess the model's learned responses of daily carbon fluxes in the challenging region of Southern Africa.

## 2 Methods

### 2.1 Datasets

Our model receives meteorological time series data for each grid-cell, including precipitation, net radiation, air temperature, vapor pressure deficit, and the atmospheric concentration of $CO_2$, as inputs (Table 1). Additionally, we incorporate static inputs, which vary only between grid-cells that consist of soil properties, land cover fractions, elevation, and wetlands, as described in (Baghirov et al., 2025).

We use the term "data constraints" to refer to the observation-based data used as target variables to optimize the model. These data guide the model's behavior through its NN components. We utilize data constraints of TWS variations, SWE, ET, runoff, fAPAR, GPP, and NEE (Table 1).

All meteorological forcing and model constraints were aggregated to a 1° spatial resolution. The spatial resolutions of static inputs were aggregated to 1/30°. Meteorological forcing data are maintained in their native daily temporal resolutions, while a monthly temporal resolution is applied to the model constraints.

We use the OCO-2 LNLGIS atmospheric inversion MIP ensemble median to constrain our estimations of NEE for the period 2015-2020, which uses satellite column and in-situ $CO_2$ observations jointly. Additionally, we use CarboScope (in-situ based NEE inversion product) to constrain the long-term global average interannual variance of NEE estimations. Both OCO-2 MIP and CarboScope estimate the carbon exchange, which includes contributions from fire emissions that we do not model. Therefore, we use fire emission data from GFED (Chen et al., 2023) to subtract fire emissions from OCO-2 and CarboScope inversions. We only use the mean seasonal cycle of ET, runoff, and GPP data to constrain the seasonality of our simulations, due to caveats of representing interannual variability of these data.



**Table 1.** Datasets used: meteorological forcing, static inputs and model constraints. The resolution column shows the original resolutions.

| Name | Resolution | | Data | Reference |
| | Spatial | Temporal | | |
|---|---|---|---|---|
| **Meteorological forcing** | | | | |
| Precipitation | 1° | Daily | GPCP 1dd v1.2 | Huffman et al. (2016) |
| Net radiation | 1° | Daily | CERES SYN1deg Ed4A | Wielicki et al. (1996), Doelling (2017) |
| Air temperature | 0.5° | Daily | CRUNCEP v8 | Harris et al. (2014), Viovy (2018) |
| Vapor pressure deficit | 1° | Daily | ERA5 | Hersbach et al. (2020), Soci et al. (2024) |
| Atmospheric $CO_2$ | 1° | Hourly | CAMS | Inness et al. (2019), Agustí-Panareda et al. (2023) |
| **Static data** | | | | |
| Soil properties | 1/120° | - | Soil grids v2 | Hengl et al. (2017), Poggio et al. (2021) |
| Land cover fractions | 1/360° | - | Globland30 v1 | Chen et al. (2015) |
| Digital elevation model | 1/120° | - | GTOPO | Center (1997) |
| Wetlands | 1/240° | - | Tootchi | Tootchi et al. (2019) |
| **Model constraints** | | | | |
| Terrestrial water storage | 0.5° | Monthly | GRACE Tellus JPL RL06M v1 | Watkins et al. (2015) |
| fAPAR | 500$m$ | 8 daily | MOD15A2H | Myneni et al. (2015) |
| Snow water equivalent | 0.25° | Monthly | GlobSnow v3 bias corrected | Luojus et al. (2014), Luojus et al. (2021) |
| Evapotranspiration | 0.05° | Hourly | FLUXCOM-X-BASE | Jung et al. (2019), Nelson et al. (2024) |
| Runoff | 0.5° | Monthly | GRUN v2 | Ghiggi et al. (2021a), Ghiggi et al. (2021b) |
| Gross primary productivity | 0.05° | Hourly | FLUXCOM-X-BASE | Jung et al. (2020), Nelson et al. (2024) |
| Net ecosystem exchange (short-term) | 1° | Monthly | OCO-2 v10 MIP | Byrne et al. (2022) |
| Net ecosystem exchange (long-term) | 1° | Monthly | Jena CarboScope | Rödenbeck et al. (2018) |

Additionally, we apply a soft constraint on the spatial distribution of mean annual CUE (NPP / GPP), based on the TRENDY

v11 ensemble of global process-oriented ecosystem models (Sitch et al., 2024). This helps to constrain plausible relative contributions of autotrophic and heterotrophic respiration in H2CM. Note that this step only constrains the magnitude of mean respiration components based on theoretical expectations, while temporal variations of CUE emerge from training H2CM.

## 2.2 H2CM

### 2.2.1 Hydrological cycle

We utilize the hydrological cycle model component of H2MV (Baghirov et al., 2025) within our model. This hydrological model estimates three primary water storages: snow, soil moisture, and groundwater. It estimates key water fluxes, including




snowfall and snowmelt, soil and groundwater recharge, evapotranspiration (divided into transpiration, soil evaporation, and interception evaporation), and runoff (divided into fast and slow runoff) to update these water storages.

In the equations below, parameters with the superscript $< s, t >$ indicate variables that vary both spatially ($s$) and temporally ($t$), while those with the superscript $< s >$ refer solely to spatial variation. The Greek letter $\beta$ represents globally constant parameters that are also learned by the neural network.

The main coupling mechanism between water and carbon cycles in H2CM is through linking transpiration and GPP by modelling WUE.

Transpiration is modeled as described by Baghirov et al. (2025):

$$T^{<s,t>} = \text{fAPAR}^{<s,t>} \cdot ET_{pot}^{<s,t>} \cdot \alpha_T^{<s,t>} \qquad \left(\text{in } \text{mm day}^{-1}\right) \quad , \qquad (1)$$

where transpiration is a function of the predicted vegetation state (fAPAR), potential evapotranspiration, and a neural network-learned parameter $\alpha_T$, which represents effective conductance or stress. The predictions of $\alpha_T$ depend on relative soil moisture, vapor pressure deficit, net radiation, and static variables (Table 2).

Further details on the hydrological model are available in Baghirov et al. (2025), and thus are not discussed here.

### 2.2.2 Carbon cycle

We employ a simple, conceptual carbon cycle model to estimate carbon cycle fluxes (Fig. 1).

GPP is the amount of carbon dioxide absorbed by vegetation to produce organic matter through photosynthesis. We model GPP as:

$$GPP^{<s,\,t>} = T^{<s,\,t>} \cdot WUE^{<s,\,t>} \cdot CO_2^{<s,\,t>} \cdot \beta_{CO_2} \qquad \left(\text{in } \text{gC m}^{-2}\,\text{day}^{-1}\right) \qquad (2)$$

where GPP is a function of transpiration $T$, atmospheric $CO_2$ concentration, the global constant $\beta_{CO_2}$ describing the $CO_2$ fertilization effect and NN learned WUE. WUE is a function of relative soil moisture, vapor pressure deficit, net radiation, and static variables (Table 2).

NPP is the net carbon available after autotrophic plant respiration. NPP is modeled as:

$$NPP^{<s,\,t>} = GPP^{<s,\,t>} \cdot CUE^{<s,\,t>} \qquad \left(\text{in } \text{gC m}^{-2}\,\text{day}^{-1}\right) \qquad (3)$$

where NPP is a function of GPP and CUE. CUE, directly learned by the NN which receives net radiation, air temperature, vapor pressure deficit, atmospheric $CO_2$ concentration, relative soil moisture, fAPAR and NPP at previous time step, and static variables (Table 2). Autotrophic respiration (Ra) is simply the difference between carbon dioxide uptake from the atmosphere (GPP) and the net carbon retained for growth (NPP).

Heterotrophic respiration, Rh, refers to the process by which non-photosynthetic organisms (e.g., microorganisms) decompose organic matter, releasing carbon dioxide back to the atmosphere. We model Rh using a traditional $Q_{10}$ function:

$$Rh^{<s,\,t>} = Q_{10}^{\frac{T_{\text{air}}^{<s,\,t>} - T_{\text{ref}}}{10}} \cdot Rb^{<s,\,t>} \qquad \left(\text{in } \text{gC m}^{-2}\,\text{day}^{-1}\right) \qquad (4)$$





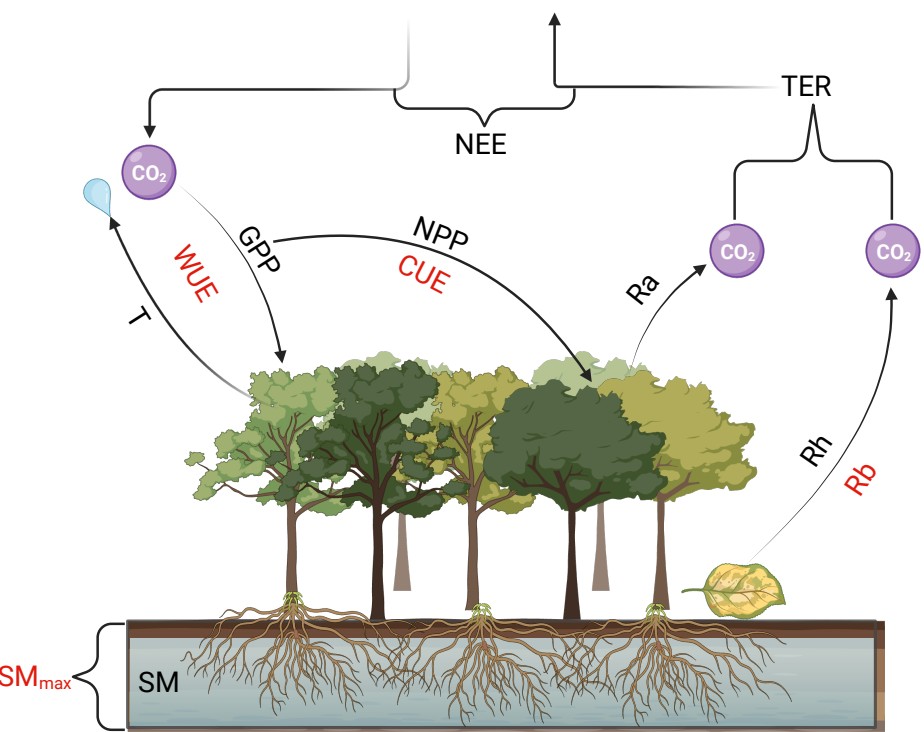

**Figure 1.** High level overview of the carbon cycle related processes in H2CM: Acronyms above or below the arrows shown in black (e.g., GPP) represent modeled parameters. Acronyms next to the arrows and modelled parameters shown in red (e.g., WUE), indicate parameters directly learned by neural networks to model the corresponding parameter next to the arrow. T: transpiration, WUE: water use efficiency, GPP: gross primary productivity, CUE: carbon use efficiency, NPP: net primary productivity, NEE: net ecosystem exchange, Ra: autotrophic respiration, Rh: heterotrophic respiration, Rb: basal respiration rate, TER: terrestrial ecosystem respiration, $SM_{max}$: maximum soil moisture capacity, SM: soil moisture. Created in BioRender (Baghirov, 2025a).

where $Q_{10}$, a global constant learned by the NN, describes the temperature sensitivity factor, indicating by which factor respiration increases with a $10°C$ rise in air temperature. $T_{air}$ is the current air temperature at time $t$, $T_{ref}$ is the reference temperature set to $15°$, and Rb is the basal respiration rate essentially representing the availability of carbon for Rh and it is learned by recurrent (stateful) NNs as a function of net radiation, precipitation, fAPAR and NPP at previous time step, and static variables (Table 2).

Terrestrial ecosystem respiration (TER) is simply defined as the sum between autotrophic (Ra) and heterotrophic (Rh) respiration representing the total amount of carbon dioxide being respired to the atmosphere.

Net ecosystem productivity (NEP) represents the amount of carbon dioxide fixed by vegetation (GPP), minus the total carbon dioxide respired to the atmosphere (TER). It indicates whether the ecosystem is gaining or losing carbon dioxide. For example,





if GPP is greater than TER (NEP > 0), the ecosystem acts as a carbon sink. Conversely, if TER is greater than GPP (NEP < 0), the ecosystem functions as a carbon source.

NEE is simply negative of NEP. NEE represents the net carbon exchange between an ecosystem and the atmosphere.

### 2.2.3 High-level overview of the hybrid architecture

H2CM consists of three primary modules: the dynamic (recurrent), the static (fully connected) and the process-based (water-carbon cycle) module:

In the dynamic module, meteorological forcing data (inputs) at a single time step $t$ (Fig. 2A) are processed through a dynamic NN layer, specifically an LSTM (Hochreiter and Schmidhuber, 1997) (Fig. 2B). LSTM is a type of recurrent neural network (RNN) architecture designed to effectively capture long-range dependencies and patterns in sequential data by leveraging its

gating mechanisms to learn memory effects (Staudemeyer and Morris, 2019). The LSTM transforms meteorological inputs at time step $t$ into higher-dimensional parameters, which are then fed into fully connected neural networks (FCNN) (this occurs within Fig. 2B but is not explicitly visualised for simplicity) to convert them into physically interpretable parameters (Fig. 2C), referred to as temporal direct predictions in this study. Additionally, the LSTM receives compressed representations of static inputs (Fig. 2F) from the static module (Fig. 2G, see also below), estimated water and vegetation states (Fig. 2E), and its

learned internal hidden and cell states (representing long-term memory) from time step $t-1$ to make estimations at time step $t$. All estimations within the dynamic module vary in both space and time.

H2CM's static module consists of an FCNN (Fig. 2G) that receives static inputs (Fig. 2F) and compress them. These compressed representations of the static inputs are then sent as inputs to the LSTM in the dynamic module (Fig. 2B). Additionally, these compressed representations are connected to another FCNN (this occurs within Fig. 2G but was omitted in the figure for

simplicity), which further transforms them into physically interpretable static parameters (e.g., $SM_{max}$, the maximum water holding capacity of the soil) (Fig. 2H). We refer to these parameters as direct static predictions.

H2CM also includes globally constant learned parameters (Fig. 2I) to represent parameters that are fixed across both space and time.

In the process-based module, meteorological forcing (Fig. 2A), temporal direct predictions (Fig. 2C), static direct predictions

(Fig. 2H), and global constants (Fig. 2I) are input into a global coupled water-carbon cycle process-based model (Fig. 2D) to estimate water and carbon fluxes, as well as water states (Fig. 2E) — collectively termed hybrid predictions. Some of these hybrid and direct predictions are directly constrained using observations or observation-based data (Fig. 2E).

When we integrate the carbon cycle with the hydrological cycle from H2MV, the number of directly estimated parameters and inputs to the model increases. This expansion raises the risk of the NN component making estimations based on unrea-

165 sonable relationships, known as short-cut learning (Geirhos et al., 2020). This issue arises due to potential interdependencies among the inputs, where different inputs or their combinations might be used to estimate certain parameters, even if such estimations are implausible based on established process knowledge.

To address this, we 'guide' NNs by ensuring they learn from plausible inputs when estimating specific parameters. Instead of employing a single large NN to predict all parameters simultaneously, which is a common approach, we divide the neural





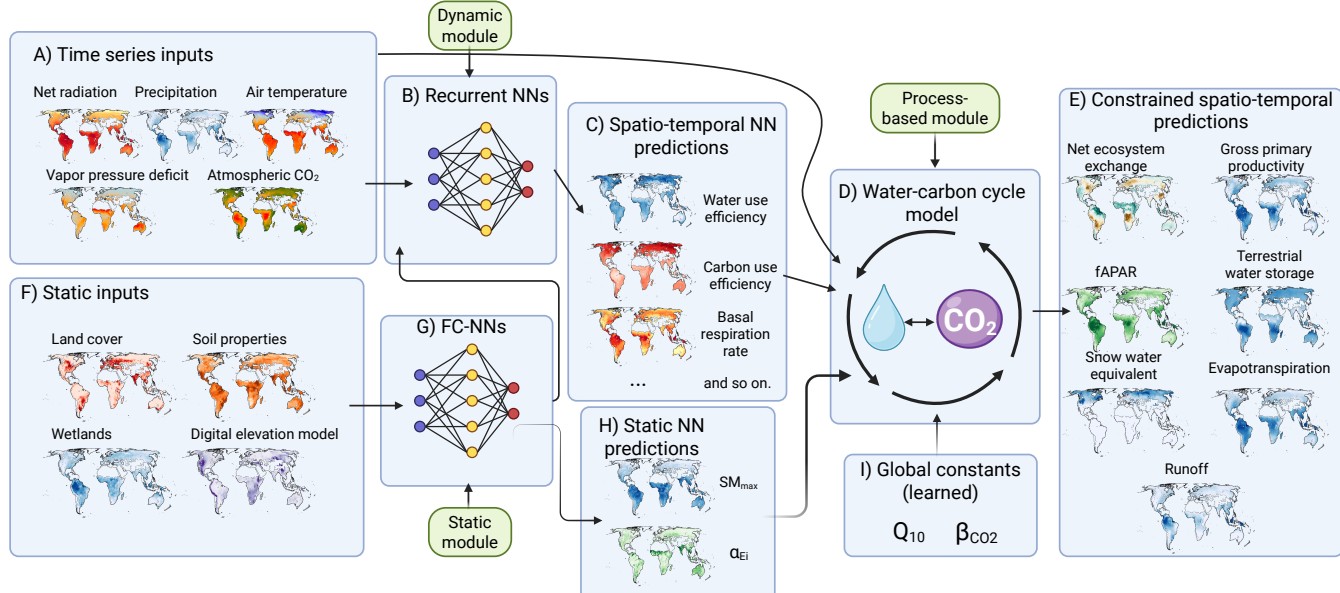

**Figure 2.** Simplified overview of the H2CM framework. Note: NNs refers to neural networks, and FC-NN refers to fully-connected neural networks. While H2CM uses LSTM as the recurrent neural network, a fully-connected neural network is illustrated here for simplicity. Note that this figure omits the 'guided' nature of the model (different groups of neural networks that recieve different inputs for their task) and only shows very high-level overview. Created in BioRender (Baghirov, 2025d).

networks into groups. Each group is restricted to learning from plausible relationships, enhancing the reliability of the model's predictions (Table 2).

## 2.3 Model optimization

### 2.3.1 Cross validation (CV)

To evaluate the generalizability of H2CM, we use a 10-fold cross-validation (CV) setup. This process involves training 10 different models, each with unique training and validation sets. The training set is used to train the model, while the validation set is exclusively for hyperparameter tuning, such as selecting the learning rate, and for early stopping. Furthermore, we keep a separate set of grids as a testing set, which is used only after training to assess the model's performance and generalizability. Importantly, the testing set is never exposed to the models during the training phase (Fig. A1).

We divide the global data into training, validation, and testing sets using a spatial-only splitting method (Fig. A1). This approach ensures that different grids are included in each of the training, validation, and testing sets. To address spatial autocorrelation among grid cells, we implement a spatial blocking technique as recommended by Roberts et al. (2017). Each block's size is 5°×5° grids, encompassing 25 grids in total. These blocks are randomly selected from the global data, ensuring



**Table 2.** Guided neural networks: The first column shows the estimated parameters, the second column lists the inputs used in the estimation process, the third column details the types of neural networks used for estimating these parameters, and the last column shows behaviour each neural network controls. WUE: Water use efficiency, CUE: Carbon use efficiency, fAPAR: fraction of absorbed photosynthetically active radiation, Rb: Basal respiration rate, FCNN: Fully connected neural networks, LSTM: Long short-term memory, Rn: Net radiation, $\frac{SM}{SM_{\max}}$: relative soil moisture, SWE: snow water equivalent, GW: Groundwater, VPD: vapor pressure deficit, $CO_2$: atmospheric carbon-dioxide concentration, NPP: Net primary productivity. The $\alpha$ parameters that are used in hydrological cycle are introduced in Baghirov et al. (2025).

| Parameter | Inputs | Neural networks | Neural network's role |
|---|---|---|---|
| $SM_{\max}, \alpha_{Ei}$ | Static | FCNN | Spatial parameters |
| $\alpha_{r_{soil}}, \alpha_{r_{gw}}, \alpha_{s_{melt}}$ | Rn, Precipitation, $\frac{SM}{SM_{\max}}$, $SWE^{t-1}$, $GW^{t-1}$, fAPAR$^{t-1}$, Static | LSTM | Most hydrological behaviour |
| $WUE, \alpha_T$ | Rn, VPD, $\frac{SM}{SM_{\max}}$, Static | FCNN | Water-carbon coupling |
| CUE, fAPAR | Rn, $T_{air}$, VPD, $CO_2$, $\frac{SM}{SM_{\max}}$, fAPAR$^{t-1}$, $NPP^{t-1}$, $static$ | LSTM | Vegetation & its respiration |
| Rb, $\alpha_{Es}$ | Rn, Precipitation, fAPAR$^{t-1}$, $NPP^{t-1}$, Static | LSTM | Upper soil sensitivity |

that the majority of the data (approximately 80% of the grids) is allocated to the training set, while the remaining 20% is evenly split between the validation and testing sets (10% each). The random selection of blocks also ensures that the training, validation, and testing sets approximately represent grids from all continents.

This split is crucial for accurately testing the model's generalizability on unseen testing grids and helps in understanding potential overfitting to the training data. Ideally, testing on both spatial and temporal dimensions, known as spatio-temporal splitting, would provide a more rigorous assessment. However, with the current set-up implementing proper time splitting for cross-validation is not feasible due to inconsistent and limited temporal coverage across different data constraints.

### 2.3.2 Loss function

We use the mean squared error (MSE) as the loss function, following the approach in Baghirov et al. (2025). The loss function MSE

$$L(X, \Theta, \beta) = \frac{1}{N_c} \sum_{c=1}^{C} \sum_{i=1}^{N_c} (y_{c,i} - \hat{y}_{c,i})^2 \tag{5}$$

evaluates the model's performance based on the inputs $X$, the neural network weights $\Theta$, and the global constants $\beta$. In this context, $C$ denotes the number of data constraints, $N_c$ is the number of data points for each constraint $c$, and $y_{c,i}$ and $\hat{y}_{c,i}$ represent the observed and predicted values for the constraint $c$, respectively. Throughout the training process, $\Theta$ and $\beta$ are adjusted to minimize the overall loss $L$.

The loss function is applied to grid cells for each data constraint and corresponding model estimation, except for the long-term NEE interannual variance (CarboScope). For this, we first calculate the average across all grid cells in each batch for both CarboScope and H2CM's model outputs. This results in a single time series representing the average long-term NEE




interannual variance for both estimations and CarboScope. The loss function is then applied to this batch-level averaged data. This effectively forces the model to fit to the global average, not to each individual grid for long-term interannual variability of NEE.

Note that we use a Z transformation on predictions and observations before computing the loss to remove the effect of
205 different units and to balance contributions of each data constraint to the total loss.

### 2.3.3 Model training

We use the strategy outlined by Baghirov et al. (2025) to optimize H2CM for each CV fold separately, which includes utilizing the Adam optimizer (Kingma and Ba, 2014). During training, all inputs to the neural networks are standardized using Z-transformation. We also implement early stopping, a technique that monitors both validation and training losses. This approach
halts further training if the model's performance on the validation set either declines or remains unchanged, thereby preventing overfitting to the training data. Throughout training, we keep track of the validation loss and select the final model based on the smallest total loss observed on the validation set. This selected model is then used for making final predictions, such as on the testing set.

### 2.4 Model evaluation

We assess our model's performance by utilizing the root mean square error (RMSE), Pearson's correlation coefficient (r), and the standard deviation ratio (SDR), which compares the predicted standard deviation to the observed standard deviation.

The mean seasonal cycle (MSC) refers to the predictable and recurring changes that occur throughout the year. We define MSC as

$$MSC\left(m\right) \;=\; \frac{1}{Y} \sum_{y=1}^{Y} p_{m,y} \tag{6}$$

where $p_{m,y}$ denotes the predicted or observed variable for month $m$ and year $y$, and $Y$ represents the total number of years.

Interannual variability (IAV) refers to the variations in a certain parameter that occur from one year to another. IAV is defined as

$$IAV\left(m,\,y\right) \;=\; p_{m,y} \;-\; MSC\left(m\right) \tag{7}$$

where $p_{m,y}$ represents the estimated or observed parameter for a certain month $m$ and year $y$, and $MSC(m)$ denotes MSC for
that specific month $m$.

Table 3 provides detailed information about emerging global patterns that we assess, with a specific focus on those related to various efficiency properties.



**Table 3.** Emerging global patterns: parameters, their abbreviations, ratios and respective units.

| Parameter | Abbreviation | Ratio | Unit |
|---|---|---|---|
| Precipitation use efficiency | PUE | $\frac{\text{NPP}}{\text{Prec}}$ | gC kgH$_2$O$^{-1}$ |
| Water use efficiency | WUE | $\frac{\text{GPP}}{\text{T}}$ | gC kgH$_2$O$^{-1}$ |
| Carbon use efficiency | CUE | $\frac{\text{NPP}}{\text{GPP}}$ | - |
| Light use efficiency | LUE | $\frac{\text{GPP}}{\text{APAR}}$ | gC MJ$^{-1}$ |

## 3   Results and discussions

### 3.1   Model evaluation

#### 3.1.1   Validation with independent test set

In this section, we evaluate the performance of H2CM in reproducing the monthly, MSC, IAV of GPP and NEE across the testing set that was not exposed to the model during training (Fig. A1).

Figure 3 illustrates the model's performance using various metrics. H2CM nearly perfectly reproduces the monthly and seasonal patterns of both GPP and NEE, with a Pearson's correlation coefficient (r) close to 1. The IAV of GPP and NEE (based on OCO-2 data) has a Pearson's r of 0.7 (median across members), while the long-term IAV of NEE has a Pearson's r of approximately 0.5. Note the small RMSE for NEE IAV, indicating that the reduced IAV variance contributes to lower correlations. Uncertainties in the inversions of NEE IAV when locating in space and time also contribute to lowered performance metrics, as some of the variance in the inversions is noise, not signal.

The SDR for the monthly and seasonal patterns of GPP and NEE is close to 1, indicating the model's strong capability in accurately reproducing these patterns. The SDR for GPP IAV is greater than one, suggesting an overestimation of GPP IAV compared to FLUXCOM-X-BASE. This outcome is expected, as we do not explicitly constrain GPP IAV, and FLUXCOM-X-BASE is known to underestimate the GPP interannual variance (Nelson et al., 2024). Conversely, H2CM underestimates NEE IAV in both short-term (OCO-2) and long-term (CarboScope) contexts, as indicated by a low SDR value.

In terms of RMSE, H2CM tends to exhibit higher errors for monthly data, followed by seasonal data, and then IAV.

H2CM effectively captures the monthly and seasonal patterns of GPP as captured by FLUXCOM-X-BASE (Fig. B1a and Fig. B1b). Although the model is only constrained by the seasonal patterns of GPP, its GPP's IAV aligns well with the monthly anomaly patterns of FLUXCOM-X-BASE (Fig. B1c). H2CM predicts larger IAV variance of GPP, which confirms the expected underestimation of interannual variance by FLUXCOM-X-BASE (Jung et al., 2020; Nelson et al., 2024).

For NEE, H2CM accurately replicates the monthly data and seasonal patterns observed in satellite inversions (Fig. B2a and Fig. B2b). Regarding NEE IAV, H2CM matches the general patterns captured by OCO-2 satellite inversions well, while some anomalies are not fully captured (Fig. B2c). To further assess our long-term NEE estimations between 2001 and 2019, we





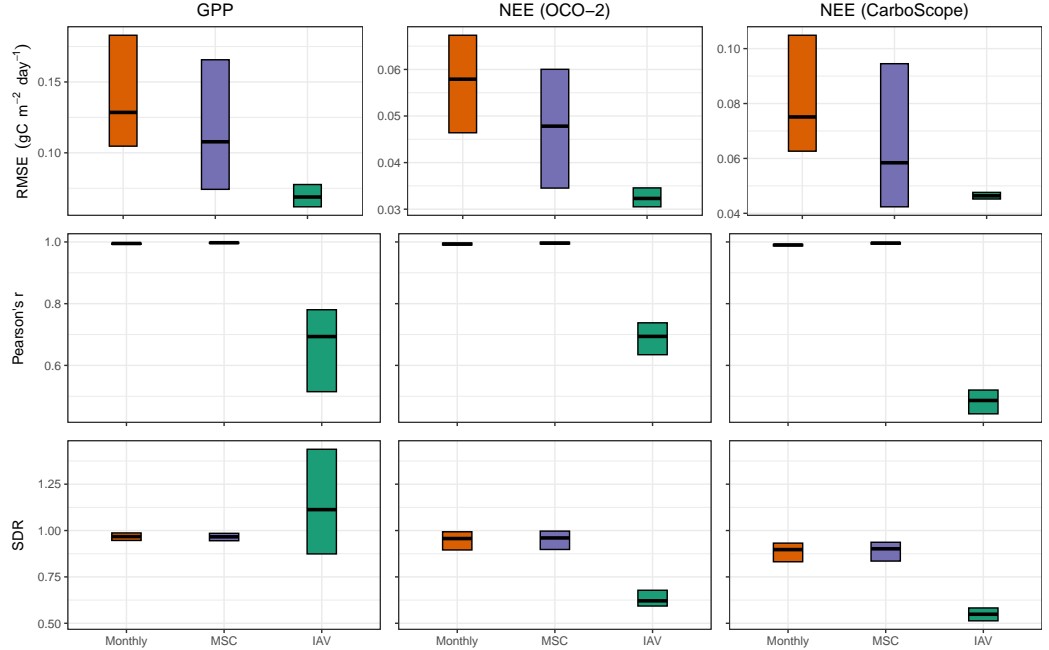

**Figure 3.** Model performance for carbon cycle constraints based on the spatial average over testing set grids. Cross bars show the maximum and minimum error, and the lines show the mean error across 10 CV folds. The rows are metrics and the columns are model constraints. RMSE refers to root mean squared error, and SDR is standard deviation ratio (the ratio between predicted and observed standard deviation).

use another inversion product, CarboScope. H2CM demonstrates strong agreement in reproducing the monthly and seasonal patterns of long-term NEE (Fig. B3a and Fig. B3b) and reasonably captures the long-term IAV of NEE (Fig. B3c).

Model performance on water cycle-related data constraints is presented in Appendix B2. We do not delve into the details
here, as the performance is qualitatively similar to that of H2MV, which has been thoroughly discussed in Baghirov et al. (2025).

### 3.1.2    Global patterns

### 3.1.3    Gross and net carbon fluxes

In this section, we evaluate H2CM's performance in terms of learned global spatial patterns. We compare H2CM's estimations
to the data constraints provided by FLUXCOM-X-BASE and OCO-2 satellite inversions, as well as to the estimations from TRENDY.

FLUXCOM-X-BASE, H2CM, and TRENDY show very similar spatial patterns of mean GPP with largest values in in wet tropical regions such as South America, Central Africa, and Southeast Asia (Fig. 4, top row). H2CM reproduces the spatial patterns of mean NEE from the OCO-2 MIP, whereas TRENDY's estimations show little spatial gradients (Fig. 4, bottom row).





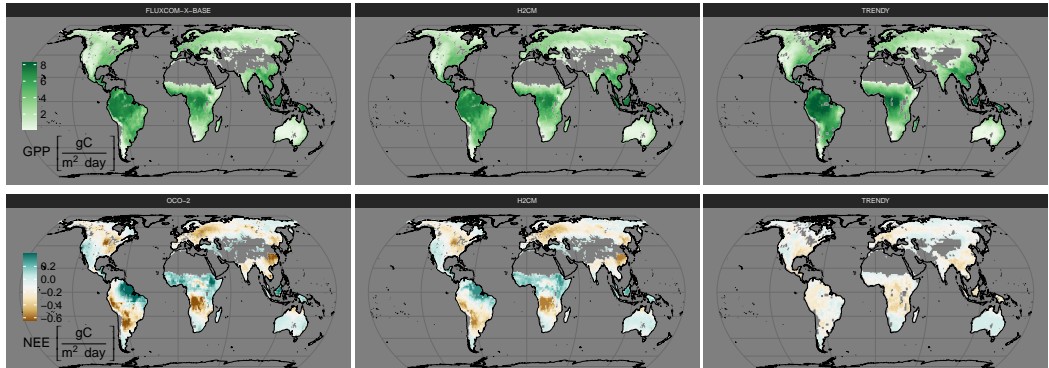

**Figure 4.** Comparison of predicted versus target global GPP and NEE against TRENDY's estimations. The figure displays the median estimations across members for H2CM (10-fold cross-validation), OCO-2 (inversion ensemble), and TRENDY (process-based model ensemble). The top row shows GPP, while the bottom row shows NEE. The mean annual average is over 2001-2019 for GPP and 2015-2019 for NEE.

Specifically, both H2CM and OCO-2 inversions suggest the largest carbon sources in northeastern South America, tropical Asia, and in the Savannah belt south of the Sarah in Africa. In contrast, the eastern parts of North America and Europe, the southwest of South America, areas south of the Equator in Africa, and the eastern parts of China appear as carbon sinks in the period 2015-2019.

### 3.1.4 Emerging global spatial patterns

H2CM estimates higher PUE values for high latitudes, particularly in regions such as western Canada, parts of Europe, and the Eurasian boreal regions (Fig. 5a). These findings strongly align with the results reported by Chen et al. (2020) that estimated rain use efficiency based on satellite data, both in terms of spatial patterns and the magnitude of the estimations. Additionally, another study by Liu et al. (2024b) employed multiple data sources, including meteorology and remote sensing, to estimate rain use efficiency, specifically focusing on Australia. Although they used GPP to estimate rain use efficiency instead of NPP

as we did, their findings for Australia are also very consistent with our model's estimates for the region. Overall, H2CM estimates low PUE in very wet and very dry regions, indicating that PUE peaks at intermediate moisture conditions. In dry areas, the low PUE pattern is attributed to vegetational constraints. Vegetation in these regions typically exhibits a low growth rate to withstand droughts, which in turn limits the response of NPP to precipitation. On the other hand, in wet regions, a large fraction of precipitation is lost to runoff, and other factors such as light and nutrients limit photosynthesis, which also causes

a decline of PUE (Paruelo et al., 1999; Huxman et al., 2004). We expect that PUE is very well constrained in H2CM because precipitation is a model input, and NPP is well constrained by data constraints on GPP and CUE.

H2CM estimates high WUE in regions such as northern North America (specifically Alaska and Canada), Northern Europe, the Eurasian boreal regions, certain areas in South America, Central Africa, the southernmost parts of Australia, and New Zealand (Fig. 5b). In contrast, H2CM predicts lower WUE for most arid regions. Low WUE estimations for arid regions is



consistent with theory and observations predicting declining WUE with increasing VPD (Boese et al., 2019; Li et al., 2023). The attenuated WUE in tropical forests might be related to larger costs of transporting water through tall trees (Prentice et al., 2013). Our study defines WUE with respect to transpiration, while empirical studies often define it as a function of total evapotranspiration. This can lead to significant differences due to spatial variations of transpiration versus evapotranspiration (Lawrence et al., 2007; Wei et al., 2017b). However, Ito and Inatomi (2012) provides global WUE estimations as a function

of GPP and transpiration based on model simulations, which align well with H2CM's WUE estimations in terms of both spatial patterns and magnitudes. H2CM's WUE is constrained by GPP and ET data, while fAPAR and the respective process formulations constrain the partitioning of ET into transpiration and evaporation components.

H2CM generally estimates higher CUE values in higher latitudes and lower CUE values in most of the Southern Hemisphere (Fig. 5c). The range of these estimations is relatively narrow, approximately between 0.45 and 0.6, indicating that roughly

45-60% of GPP is converted into net biomass. The patterns reflect the general expectations that the fraction of autotrophic respiration increases with temperature and biomass. This is expected because we have constrained the spatial pattern of mean annual CUE to align with predictions from the TRENDY ensemble of process models. Other studies have consistently reported similar findings in terms of magnitude and spatial patterns (He et al., 2018; Konings et al., 2019; Liu et al., 2019; Tao et al., 2023; Wang et al., 2022; Zhang et al., 2014).

H2CM estimates high LUE for regions such as the northwest parts of North America, the eastern states of Canada and the United States, Europe, the Eurasian boreal regions, South America, Central Africa, and Southeast Asia (Fig. 5d). Low LUE for dry regions are expected due to increased water limitations. A study by Liu et al. (2024a), using FLUXNET site data and satellite-derived proxies, found results closely matching ours, particularly in spatial patterns. Similarly, Wei et al. (2017a) used machine learning to estimate LUE, aligning mostly with our estimations, though they report lower LUE values for the

northwest part of North America and the Eurasian boreal regions. This discrepancy across different studies in some regions indicates uncertainty in estimating LUE. It may result from variations in methods, datasets, and underlying assumptions used. LUE is well constrained in H2CM by data constraints of GPP and fAPAR.

### 3.1.5 Emerging global constants

For $Q_{10}$ (unitless) the median value learned across CV folds is 1.24. The values range from 1.2 to 1.28, indicating high

robustness across the 10-fold CV folds. However, previous studies have reported $Q_{10}$ value for heterotrophic or soil respiration to fall between 1.4 and 2 (Zhou et al., 2009; Meyer et al., 2018; Niu et al., 2021), which are slightly higher than H2CM's estimations.

For $\beta_{CO_2}$ the median value learned across CV folds is 40 (% 100 ppm$^{-1}$). The range of values extends from 21.12 (% 100 ppm$^{-1}$) to 52.45 (% 100 ppm$^{-1}$), suggesting higher variability and thus lower robustness in these predictions across the 10-CV folds.

Previous studies have reported significantly smaller $\beta_{CO_2}$ values than H2CM's estimates. For example, $\beta_{CO_2}$ estimations based on observations have been reported to be approximately between around 14 and 16 % 100 ppm$^{-1}$ (AINSWORTH and ROGERS, 2007; Wang et al., 2020; Ueyama et al., 2020; Zhan et al., 2024).



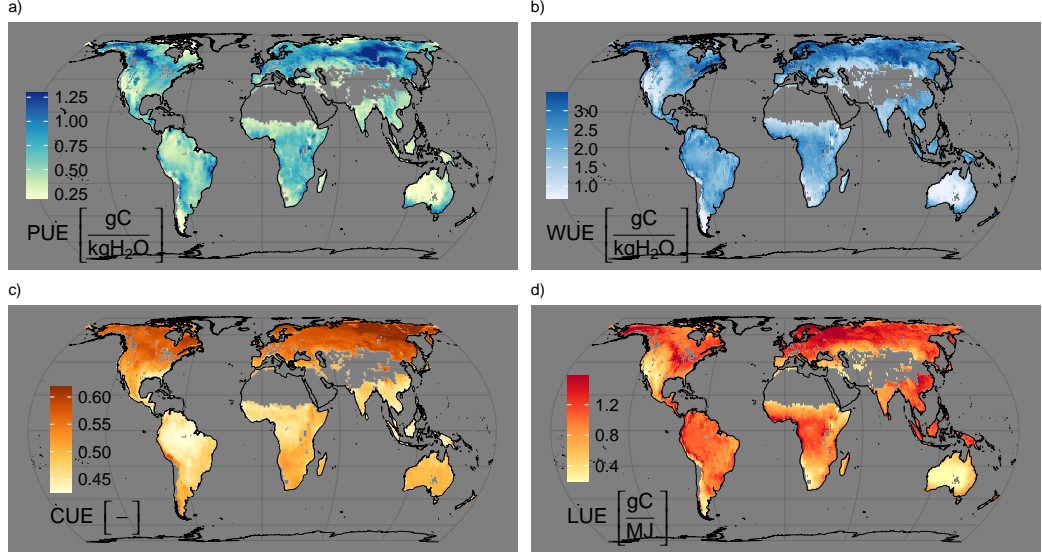

**Figure 5.** Emerging global patterns: a) Precipitation Use Efficiency (PUE), b) Water Use Efficiency (WUE), c) Carbon Use Efficiency (CUE), and d) Light Use Efficiency (LUE). The maps represent the median across 10 cross-validation folds. The mean annual average is calculated over the years 2001-2019.

This discrepancy between H2CM's global constants and the estimations from previous studies could be attributed to potential equifinality within H2CM's NN learned parameters. For example, H2CM's global constants may not be fully identifiable and could be significantly influenced by compensation among other NN learned parameters.

To address this issue, we conduct an additional experiment in which we place priors on the learned global constants (Appendix C). In this setup, H2CM estimates $\beta_{CO_2}$ to be 15 (% 100 ppm$^{-1}$) and $Q_{10}$ to be 1.5. However, these values match the priors exactly, indicating that H2CM's global constants are underconstrained by the current setup—that is, by the available observational data and the applied process knowledge constraints.

## 3.2 Estimated seasonality of carbon fluxes across major regions

In this section, we assess the estimated seasonality of the main carbon fluxes as predicted by H2CM. Our evaluation focuses on global and four major TransCom regions covering different climates: Eurasian Boreal, North American Temperate, South American Tropical, and Southern Africa (Fig. D1). We chose to conduct this evaluation on a regional basis, apart from globally, because models often encounter greater challenges when estimating carbon fluxes for specific regions (Jung et al., 2020; Lee et al., 2024; Metz et al., 2025; Nelson et al., 2024; Quetin et al., 2020; Bodesheim et al., 2018).





Conceptually, H2CM represents a gradient between purely data-driven approaches and purely process-based models. There-fore, we compare our estimations with both a pure data-driven model (FLUXCOM-X-BASE, Nelson et al. (2024)) and state-of-the-art process-based models (TRENDY, Sitch et al. (2024)).

Overall, H2CM outperforms both FLUXCOM-X-BASE (in terms of NEE) and TRENDY (in terms of NEE and GPP)
globally and across all regions, as measured by RMSE and $R^2$ (Fig. 6).

H2CM, FLUXCOM-X-BASE, and TRENDY median all accurately capture most of the seasonal variation in NEE globally and within the Eurasian Boreal region. However, the range among different TRENDY models is very large, and the ensemble median underestimates the peak of net carbon uptake in summer, apparently due to an underestimation of GPP (Fig. 6).

For the North American Temperate region, both H2CM and FLUXCOM-X-BASE effectively capture most of the seasonal
variation in NEE, with $R^2$ values of 1 and 0.98, respectively. In terms of TER, there is good agreement across all models (Fig. 6). Similar as for the Eurasian boreal region, TRENDY median underestimates the net carbon uptake peak in summer peak of NEE, and exhibits a slight shift of NEE seasonality. This issue seems attributable to TRENDY's GPP estimations, where similar problems are observed (underestimation of the peak and slight shift).

In the South American Tropical region, only H2CM captures 75% of the subtle seasonal variation in NEE, which shows
a seasonal cycle characterized by a double peak of net carbon release. FLUXCOM-X-BASE captures the first carbon release peak qualitatively but not the second, while TRENDY's seasonality seems unrelated to the OCO-2 inversion estimates.

Interestingly, the seasonality of GPP is similar between H2CM, FLUXCOM, and TRENDY, while seasonality of TER deviate across models. The discrepancy in NEE variations with H2CM is dominated by TER, as H2CM estimates a positive TER peak in October, which TRENDY and FLUXCOM-X-BASE do not. The peak of net carbon uptake suggested by OCO-2
inversions does not coincide with the GPP peak, which is qualitatively reproduced by H2CM. One possible explanation for the poor NEE estimations in the South American Tropical region is the potential omission of certain processes in the models. Given the role of respiration in shaping net carbon release patterns in the wet tropics and the apparent discrepancies related to TER among models, it seems that wetness effects on respiration remain to be a modelling issue. H2CM's neural network component seems effective to compensate for this to some extent by learning responses from the observational constraints.

In Southern Africa, H2CM explains most of the variation in NEE with an $R^2$ of 0.97. TRENDY reasonably reproduces the patterns with an $R^2$ of 0.76, while FLUXCOM-X-BASE performs poorly with an $R^2$ of 0.16. Also in this subtropical dry region, NEE seasonality and discrepancies among models seem related to respiration as both H2CM and TRENDY capture most of the variation in GPP in terms of $R^2$ (Fig. 6).

H2CM accurately reproduces the prominent peak of net carbon release at the end of the year, whereas TRENDY captures
it poorly, showing a slight shift in the peak estimation, and FLUXCOM-X-BASE completely misses it. Several studies have highlighted the challenges of correctly capturing NEE seasonality in dry regions (Jung et al., 2020; Lee et al., 2024; Metz et al., 2025; Nelson et al., 2024; Bodesheim et al., 2018). The carbon release peak in Southern Africa was attributed to the rapid stimulation of microbial respiration due to re-wetting when transitioning from the dry to the rainy season (Metz et al., 2023, 2025). Also for dry subtropical regions FLUXCOM-X-BASE and TRENDY models struggle to accurately represent
the differential responses of photosynthesis and respiration processes to water effects. These carbon-water interactions lead





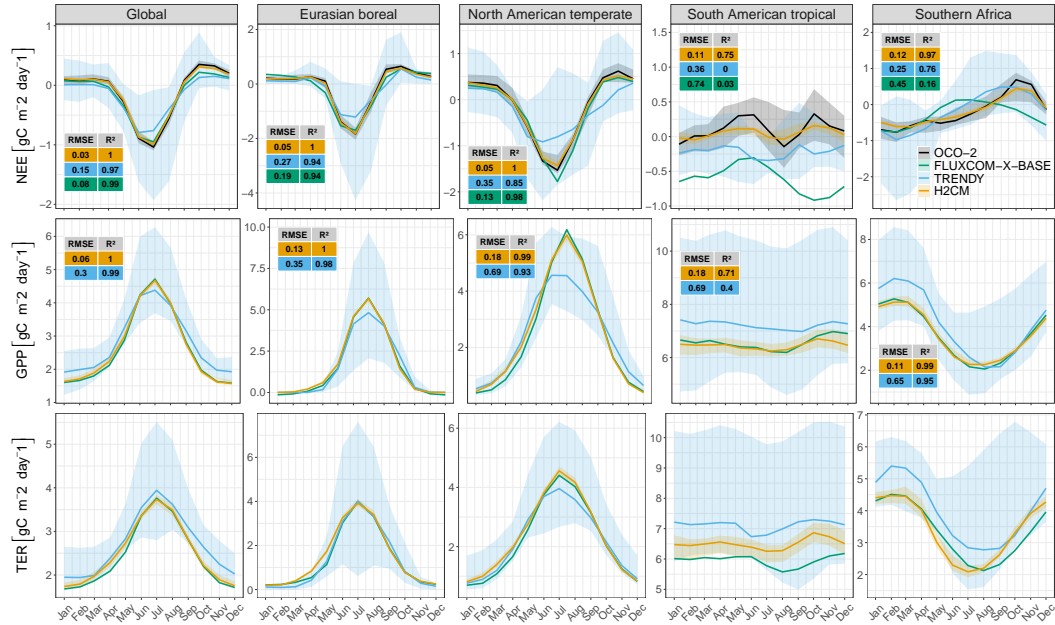

**Figure 6.** Predicted seasonality of carbon fluxes compared with data constraints, FLUXCOM-X-BASE, and TRENDY across four major Transcom regions. Shaded areas represent the range across model members (or 10 CV folds in H2CM), with lines indicating the median. Each column corresponds to a different region, and each row represents a distinct carbon flux. Subplots for NEE and GPP include tables with metrics for each region. NEE comparisons involve H2CM, FLUXCOM-X-BASE, and TRENDY against OCO-2 inversions, while GPP comparisons involve H2CM and TRENDY with FLUXCOM-X-BASE. Metrics for TER are omitted due to the absence of direct data constraints. Note varying y-axis scales for different regions.

to complex seasonal variations of NEE, which are characterized by carbon release peaks in contrast to boreal and temperate regions dominated by a GPP driven carbon uptake peak. In the next section, we further diagnose the origin of the carbon release peak in Southern Africa by H2CM.

### 3.3 Net carbon release peak in Southern Africa

To infer which regions contribute most to the end-of-year net carbon release peak in Southern Africa, we plotted the NEE difference between December and August (Fig. 7a). Interestingly, the largest amplitude occurs in the sub-humid region covering the Miombo woodlands, and not in the more dry southern part of Africa. Similarly, Metz et al. (2025) inferred that the more northern regions of Southern Africa contribute most to the peak based on an atmospheric inversion using GOSAT data.

To better understand the simulated carbon dynamics by H2CM, we plot daily variations of NEE, NPP, and Rh for a grid cell
in the center of this region (Fig. 7a). The NEE variations from August to December are characterized by a gradual increase until about November, driven by a concomitant rise in Rh. With the onset of the rain, both Rh and NEE exhibit sharp increases. This phenomenon aligns with the "Birch effect" (Birch, 1964; Jarvis et al., 2007), where re-wetting causes a respiration pulse.





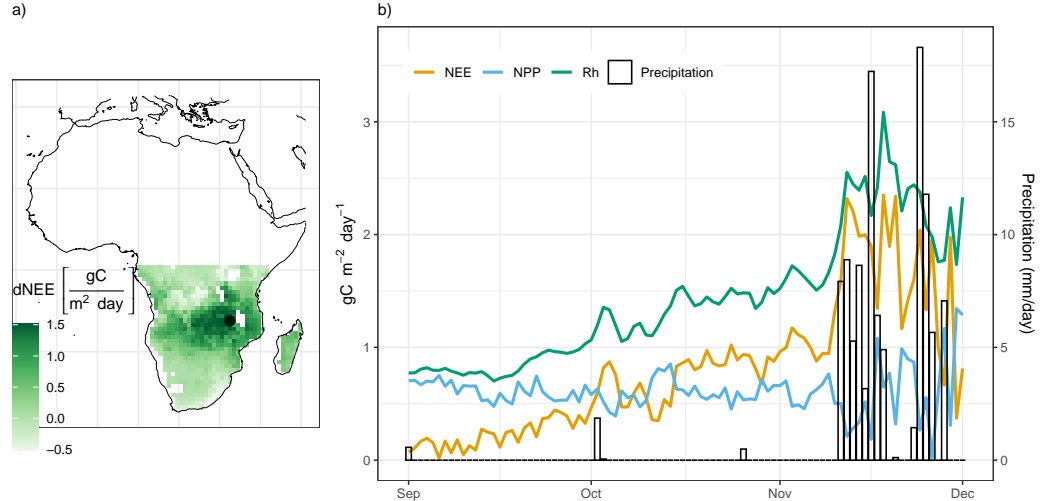

**Figure 7.** a) The difference in NEE between the peak month (December) and the onset of the gradual increase (August). The black point on the map indicates the specific grid location from which the time-series data is derived. b) Response of $CO_2$ fluxes to rainfall in Southern Africa. The left y-axis represents $CO_2$ flux values, while the right y-axis corresponds to precipitation values. The time-series is based on daily flux predictions from September through December for a single year and median across 10-CV folds.

Recent studies based on atmospheric inversions using GOSAT data have found that this effect shapes NEE variations in dry areas such as Australia (Metz et al., 2023) and Southern Africa (Metz et al., 2025).

Interestingly, at this location, we also observe that NPP declines when it rains, which further amplifies the NEE response. This suggests that in H2CM, GPP is limited by light rather than water, due to reduced radiation from cloudiness. In fact, previous studies indicate that the Miombo forests remain green throughout the dry season (Küçük et al., 2022), likely due to water access through deep rooting (Stocker et al., 2023). Our findings based on H2CM suggest that light limitation of photosynthesis in the Miombo contributes to the end-of-season NEE peak in Southern Africa, in addition to the previously

known re-wetting driven respiration response.

This example highlights H2CM's ability to capture and explain complex carbon cycle phenomena. It suggests that H2CM successfully learned carbon flux responses to high-frequency (daily) variations in environmental conditions, even though only low-frequency (monthly) data constraints were used during training.

### 3.4   Limitations

H2CM heavily depends on data-driven learning, incorporating both meteorological forcings and data constraints. The primary constraints on the carbon cycle include GPP seasonality from FLUXCOM-X-BASE, NEE from OCO-2, and global NEE IAV from CarboScope. Each of these data sources is subject to uncertainties, which could propagate to H2CM. We tried to mitigate known issues of the observational constraints by making the loss function sensitive to the more robust patterns in the data.



Our GPP constraint utilizes a machine learning approach that upscales $CO_2$ flux observations from site-level towers to global
data. A notable limitation of this method is uncertainty related to extrapolations for conditions poorly sampled by observations
(Jung et al., 2020; Nelson et al., 2024).

To constrain H2CM's NEE estimations, we utilize an atmospheric inversion ensemble using in-situ and OCO-2 satellite-
based column $XCO_2$ data. Despite significant efforts and advancements in $XCO_2$ retrievals from OCO-2, the referential
sampling of clear-sky conditions and remaining retrieval biases may affect inferred NEE variations (Connor et al., 2016;
Kulawik et al., 2019a, b; Worden et al., 2017). Additional uncertainties from various methodological choices, particularly in
atmospheric transport modeling, result in poorly constrained spatial inversion results (Yun et al., 2025; Zhang et al., 2022;
Byrne et al., 2022). We used 1° OCO-2 MIP results to constrain NEE, which, despite the added value of extensive XCO2
data, may be overly optimistic. Therefore, some discrepancies between H2CM-simulated and OCO-2 MIP-based NEE at 1°
resolution (Fig. 4, bottom row) are due to this data constraint uncertainty. Since the issue of loose spatial constraints is even
more pronounced for atmospheric inversions using only the sparse in-situ CO2 network, we used only the spatial batch average
as a data constraint from CarboScope.

H2CM can only fit the data constraints if two conditions are met: (1) the information is available in the meteorological
forcing, and (2) the model's process formulations permit it. Consequently, H2CM does not perfectly fit the data constraints,
even with its highly data-adaptive neural network component. This can, to some extent, prevent the model from adapting to
uncertainties within the data constraints (Baghirov et al., 2025).

Lastly, our model is susceptible to the equifinality problem, where different processes or pathways can lead to the same end
result, similar to process-based models (Baghirov et al., 2025). Our cross-validation (CV) setup, using 10 different models
trained on various data folds, provides some indication of how well the simulations are constrained by examining the spread
among these models.

## 415   4   Conclusions

In this study, we integrated a simple carbon cycle model with a previously developed global hybrid hydrological model that
includes vegetation dynamics. Our focus was on modeling key carbon fluxes, including GPP, NPP, TER and NEE.

We used deep learning to directly estimate critical yet uncertain carbon cycle parameters, such as water and carbon use
efficiency, and basal respiration rate. We constrained the seasonality of GPP using the observation-based upscaled data. For
NEE, we utilized satellite inversion data from 2015 to 2020, and for long-term global interannual variability of NEE, we used
a long-term inversion product based on various measurements. By constraining both GPP and NEE, we were also able to
indirectly constrain TER.

We evaluated our model's accuracy by visually and quantitatively comparing it against the data constraints, using various
metrics on a testing set that was not used during training, as well as assessing global patterns (mean annual). Our results
demonstrate that the model accurately reproduces the monthly patterns, mean seasonal cycle, interannual variability, and global
patterns (mean annual) of both GPP and NEE.



We also identify emerging global patterns in water-carbon cycle parameters, such as precipitation, water, carbon, and light use efficiency, which align with our expectations and the current literature.

We compared our model's estimations of key carbon fluxes with those from the data-driven and the state-of-the-art process-based model ensemble. Our analysis focused on different climatic regions. Our findings reveal that our model outperforms both data-driven and state-of-the-art process-based models in estimating NEE and GPP seasonality across all regions, with particularly notable performance in the South American tropics and Southern Africa. We attribute this discrepancy to water effects in wet and dry regions that are not well represented in models. However, H2CM's neural network component seems to have partially captured these processes, demonstrating the advantage of using a hybrid model when process knowledge is incomplete.

Additionally, we evaluated the estimated daily responses of carbon fluxes to sudden rainfall in a dry region, specifically focusing on the woodlands in Southern Africa. Our findings indicate that sudden rainfall in this area leads to increased NEE, driven by a rise in heterotrophic respiration, which is further amplified by reduced NPP due to light limitation. This response is implicitly learned during model training and is supported by evidence from previous studies.

H2CM opens new avenues for studying global carbon-water cycles and is a promising candidate for further development into a hybrid land surface model. The next step is to integrate the energy cycle in order to advance the development of a new generation of hybrid land surface models.

*Code and data availability.* The model simulations, aggregated to a monthly resolution, are accessible via https://doi.org/10.5281/zenodo.15785261 (Baghirov, 2025c). The initial release of the complete model code can be accessed via https://doi.org/10.5281/zenodo.15784689 (Baghirov, 2025b). For the most current version of the code, please visit the public repository at https://github.com/zavud/h2cm (last access: 1 July 2025). We are open to sharing the original daily simulations and additional variables (that are not shared) upon request.




## Appendix A: Cross validation

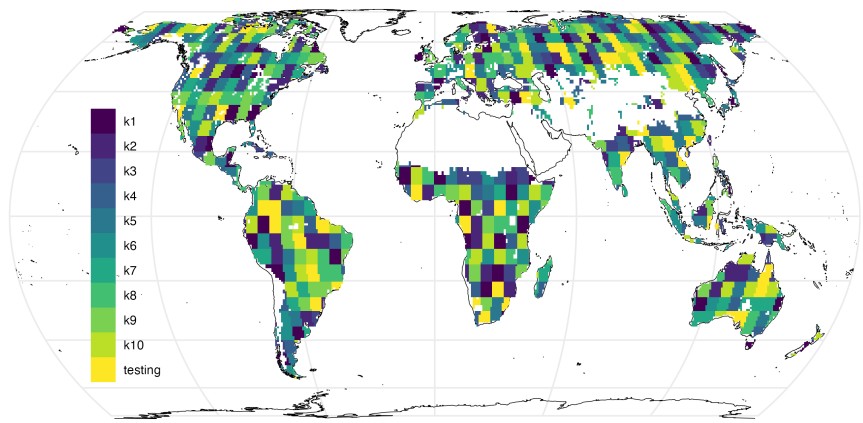

**Figure A1.** Validation sets for 10 different models and a fixed testing set. Note that, during training, each fold has a separate and unique validation set and all models were tested on the same testing set.

## Appendix B: Model evaluation

### B1    Model performance on the carbon cycle constraints

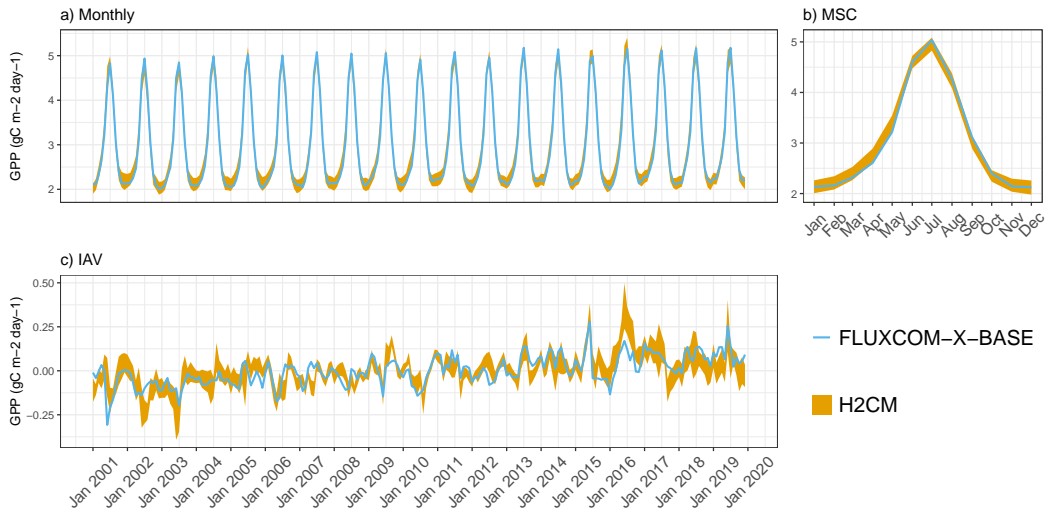

**Figure B1.** Predicted versus target GPP (FLUXCOM-X-BASE) over the testing set (spatial domain) across 10 CV folds: a) monthly, b) mean seasonal cycle and c) interannual variability




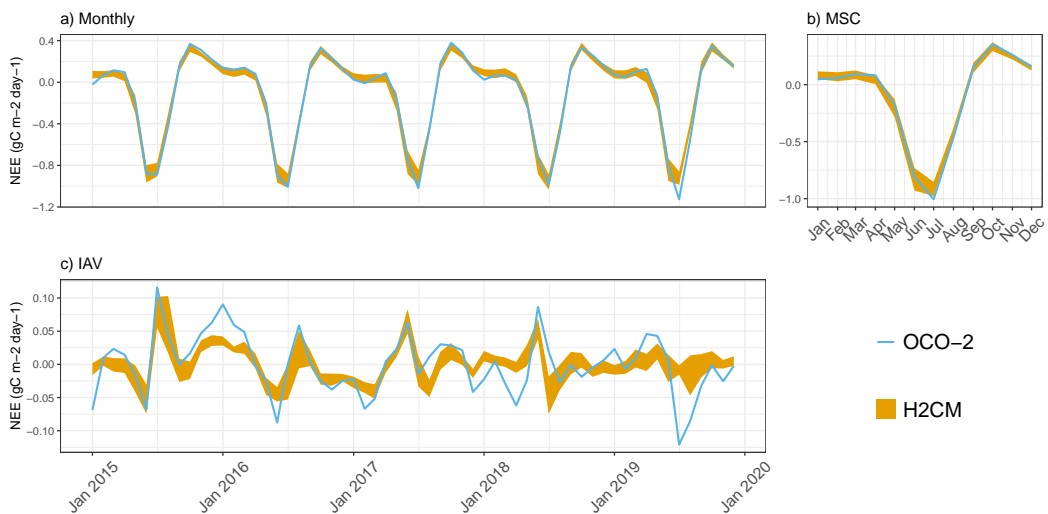

**Figure B2.** Predicted versus target NEE (OCO-2) over the testing set (spatial domain) across 10 CV folds: a) monthly, b) mean seasonal cycle and c) interannual variability

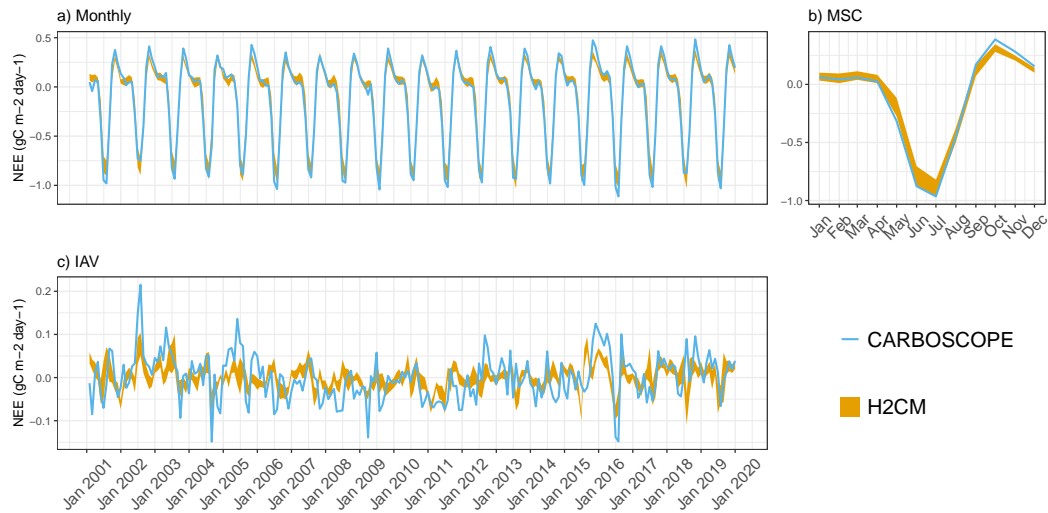

**Figure B3.** Predicted versus target NEE (CarboScope) over the testing set (spatial domain) across 10 CV folds: a) monthly, b) mean seasonal cycle and c) interannual variability



**B2    Model performance on the water cycle constraints same as in H2MV**

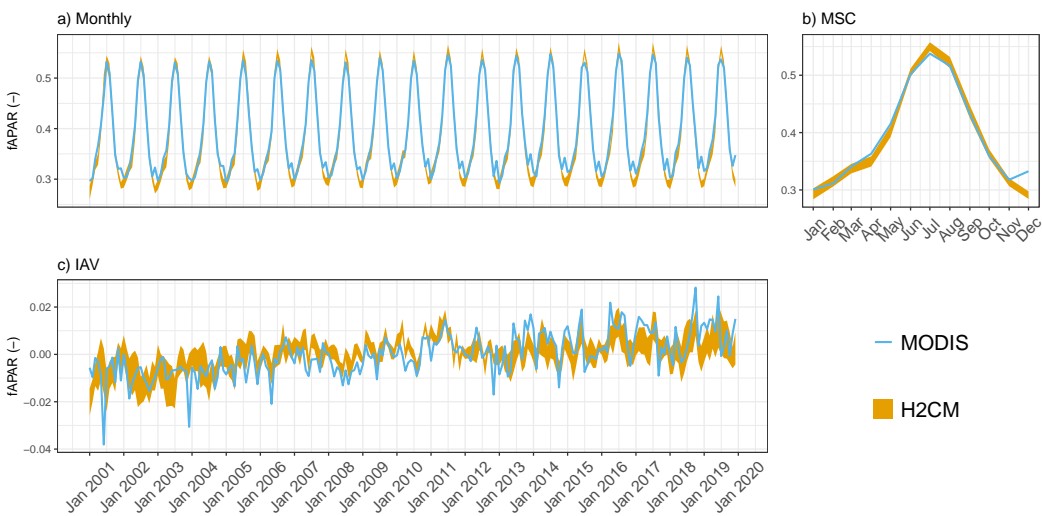

**Figure B4.** Predicted versus observed mean fAPAR over the testing set (spatial domain) across 10 CV folds: a) monthly, b) mean seasonal cycle and c) interannual variability.

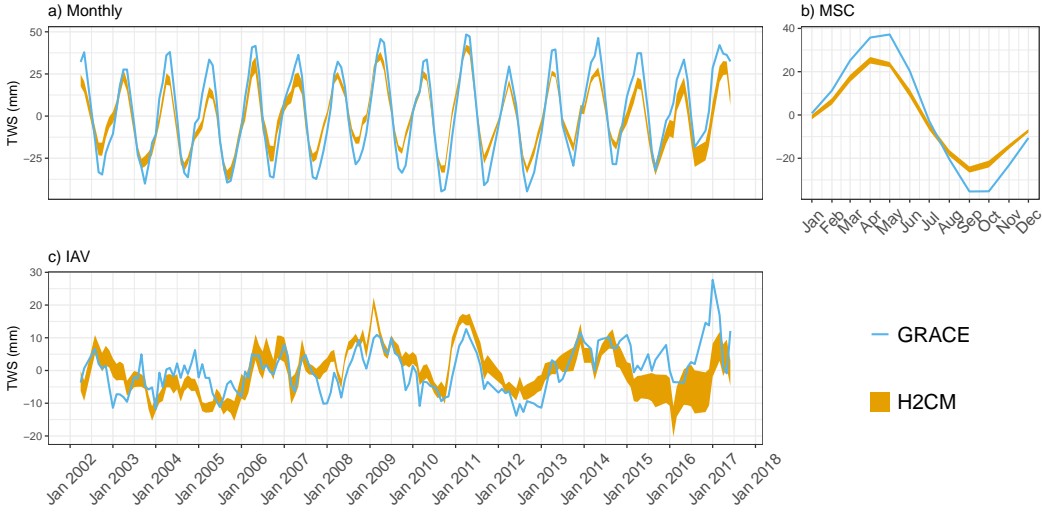

**Figure B5.** Predicted versus observed mean TWS (anomaly) over the testing set (spatial domain) across 10 CV folds: a) monthly, b) mean seasonal cycle and c) interannual variability.



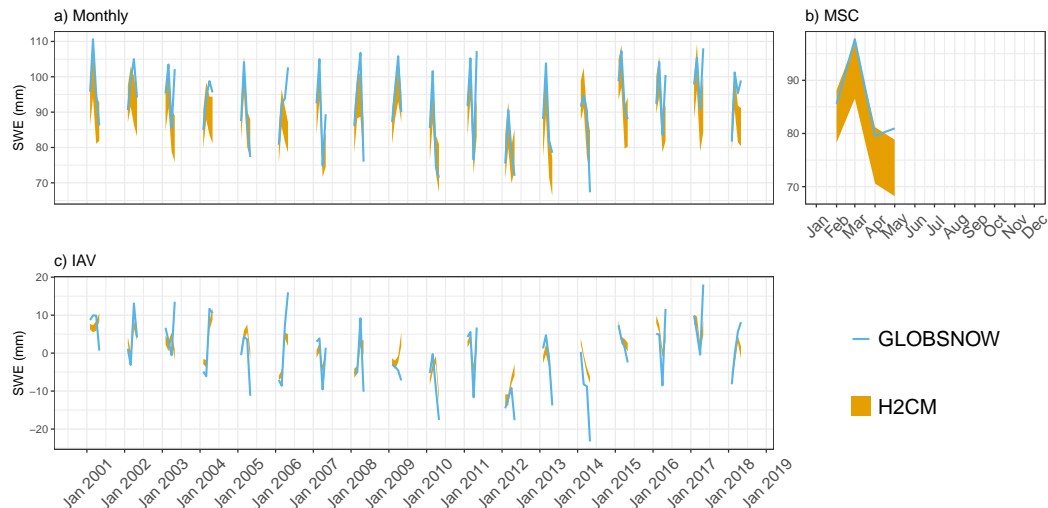

**Figure B6.** Predicted versus observed mean SWE over the testing set (spatial domain) across 10 CV folds: a) monthly, b) mean seasonal cycle and c) interannual variability

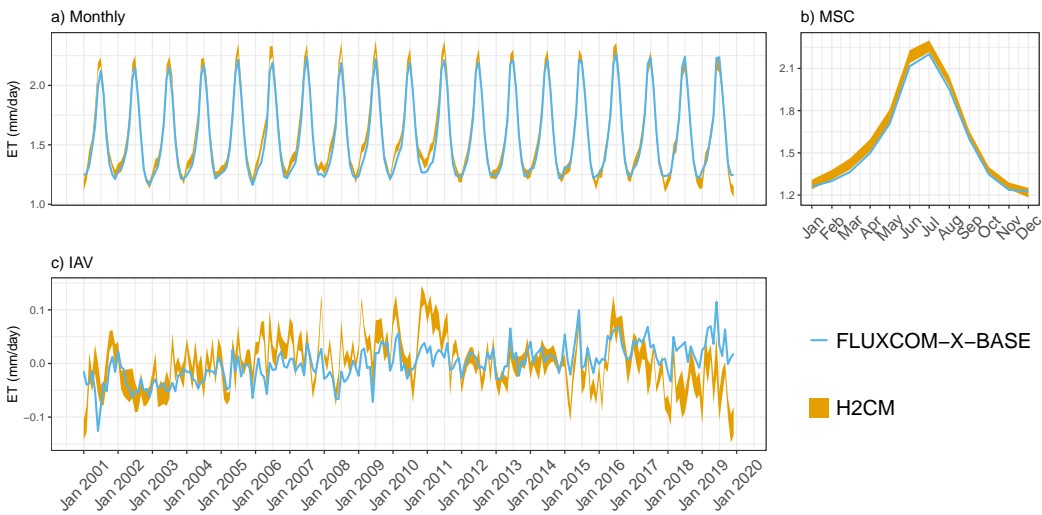

**Figure B7.** Predicted versus target mean ET over the testing set (spatial domain) across 10 CV folds: a) monthly, b) mean seasonal cycle and c) interannual variability



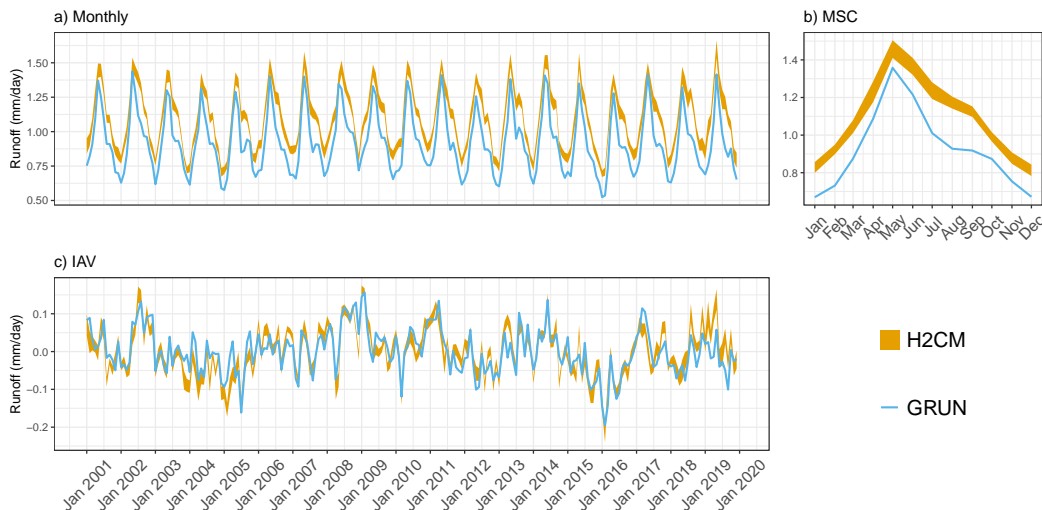

**Figure B8.** Predicted versus target mean Runoff over the testing set (spatial domain) across 10 CV folds: a) monthly, b) mean seasonal cycle and c) interannual variability.

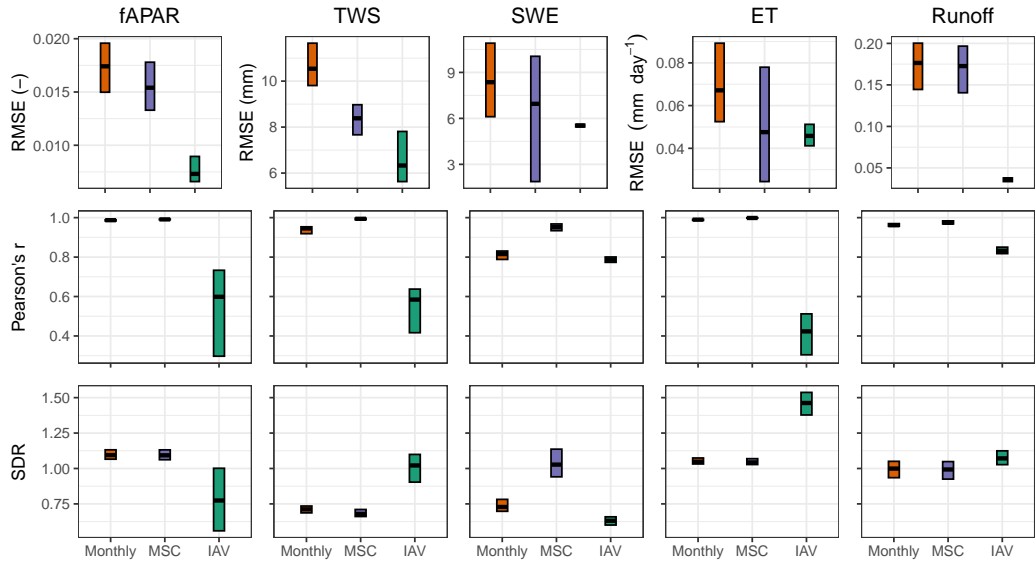

**Figure B9.** Model performance for water cycle constraints on the testing set. Cross bars show the maximum and minimum error, and the lines show the mean error across 10 folds. The rows are metrics and the columns are model constraints. RMSE refers to root mean squared error, and SDR is standard deviation ratio (the ratio between predicted and observed standard deviation).





## Appendix C: Prior penalty on global constants

To incorporate prior knowledge into H2CM's estimation of its global parameters, we add the following penalty term to the loss:

$$l_p = \frac{(\hat{\theta} - \theta)^2}{\sigma_\theta^2} \tag{C1}$$

Here,

- $\hat{\theta}$ is the estimated global constant,

- $\theta$ is its prior mean, and

- $\sigma_\theta$ is the prior standard deviation.

In our experiments:

– For $\beta_{CO_2}$, we set

$$\theta = 15\% \, 100 \, \text{ppm}^{-1}, \quad \sigma_\theta = 5\% \, 100 \, \text{ppm}^{-1}.$$

- For the temperature sensitivity, $Q_{10}$, we use

$$\theta = 1.5, \quad \sigma_\theta = 0.3.$$

Under this setup, H2CM estimates

$\beta_{CO_2} = 15\% \, 100 \, \text{ppm}^{-1}, \quad Q_{10} = 1.5,$

which coincide with their priors and indicate that, without additional data and process constraints, H2CM's global constants remain underdetermined by the current data and process knowledge constraints alone.



## Appendix D: Major transcom regions studied

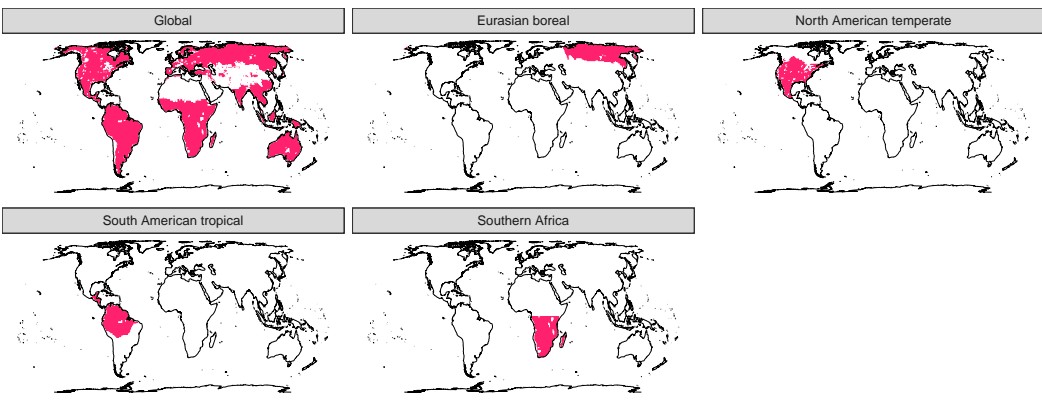

**Figure D1.** Major transcom regions studied in this study

*Author contributions.* ZB implemented the model, conducted the analysis, and drafted the manuscript. MJ designed main components of the water-carbon cycle model structure, with inputs from MR, BA, and ZB. All authors contributed intellectual input to the design, analysis, and writing."

*Competing interests.* The authors declare no competing interests.

*Acknowledgements.* We gratefully acknowledge financial support through the German Aerospace Center (DLR) with funds provided by the Federal Ministry for Economic Affairs and Climate Action (BMWK) due to an enactment of the German Bundestag. We further acknowledge the support by the European Research Council (ERC) Synergy Grant "Understanding and Modelling the Earth System with Machine Learning (USMILE)" under the Horizon 2020 research and innovation programme. Zavud Baghirov is supported by the International Max Planck Research School for Global Biogeochemical Cycles (IMPRS-gBGC). Bernhard Ahrens was funded by the European Union Horizon Europe research and innovation program under Grant agreement No. 101086179 (AI4SoilHealth). We gratefully acknowledge the financial support from the Max Planck Society, which enabled us to publish this paper as open-access.



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
