# Peer review of "H2CM (v1.0): hybrid modeling of global water–carbon cycles constrained by atmospheric and land observations"

_EGUsphere, 2025_

## Author Comment (AC3)

**Author's response to reviewers**

We appreciate the detailed reviews provided by both reviewers, which have significantly contributed to improving the clarity of our manuscript.

We display the reviewers' comments in black and italic and highlight our responses in green and changes we made in blue in order to ensure clarity.

Best regards, Zavud Baghirov on behalf of the authors

**1 Reviewer 1**

The authors present a new global hybrid model (H2CM) that couples terrestrial water and carbon cycles by blending physically based equations with neural network components, constrained by multiple observational data streams. The study is timely and potentially significant, given growing interest in machine learning augmented Earth system models. The authors clearly describe the model design, data constraints, and evaluation. The integration of a hybrid hydrological model (H2MV) with a conceptual carbon cycle model is new. The results demonstrate strong performance, notably in capturing seasonal carbon flux patterns that some process models miss. I find the work scientifically interesting and largely well executed. However, several clarifications and improvements are recommended:

Dear reviewer 1,

Thank you very much for taking the time to review our manuscript and for providing feedback that helps us improve its clarity.

**1.1 Scientific significance and novelty**

The authors propose the first global hybrid model explicitly coupling water and carbon cycles with ML-guided parameters. This addresses a recognized gap, the integration of observational constraints on both hydrology and carbon is novel. The model's ability to reveal patterns (e.g., precipitation-use efficiency, water-use efficiency) demonstrate added value beyond traditional

models. The work thus represents a significant advance toward next-generation hybrid land-surface models. I suggest that the authors should highlight more explicitly how H2CM differs from and advances prior approaches. Similarly, highlight that hybrid modeling is still "young and evolving" (l.47) and that most previous work was at the proof-of-concept stage, underscoring H2CM's novelty. If there are any other related models (even sub-global studies), a brief comparison would strengthen the novelty claim.

We thank the reviewer for this helpful suggestion.

In the introduction we now mention that:

"However, it is important to note that this is still a young and evolving field and most current works on hybrid modelling were at the proof-of-concept stage."

We also added the following paragraph to the introduction to explain how H2CM differs from and advances the most commonly used approaches for modeling global water and carbon cycle components:

"H2CM advances the field by combining previous approaches—such as fully data-driven models like FLUXCOM (Jung et al. 2019, 2020; Nelson et al. 2024) and fully process-based models like TRENDY (Sitch et al. 2015, 2024)—by aiming at combining their strengths: it learns directly from global observations through its machine learning component, while respecting conceptual process understanding and mass balance."

To the best of our knowledge, no hybrid model of the coupled water—carbon cycle existed at global or subglobal (e.g., continental) scales prior to this work; here we present the first. For framing and context relative to other emerging modelling strategies, please see our response in Section 2.1.1.

**1.2 Methodology and model design**

The model architecture is generally well described. H2CM extends H2MV hydrology by adding a carbon cycle (Eqs. 1-4). Transpiration is computed from FAPAR, potential ET, and a parameter alpha\_T (Eq. 1). GPP is linked to transpiration via a NN learned WUE and a CO2-fertilization term beta (Eq. 2). NPP uses a NN learned CUE (Eq. 3), and heterotrophic respiration (Rh) follows a Q10 function (Eq. 4) with a NN learned basal respiration rate Rb. The modeling choices are physically plausible, and the coupling (via WUE linking T and GPP) is reasonable. Table 2 clarifies how each neural network is guided by selecting meaningful inputs (e.g., WUE depends on soil moisture, VPD, radiation). This guided-NN strategy improves interpretability.

Is there one NN for each output variable? (ll.60) Why was it better to use several models? Do you performed experiments using one model for several outputs with various inputs? Please be more detailed here and explain why your approach was best.

We thank the reviewer for this question. To clarify, we revised the H2CM conceptual figure (see our response in Section 1.2.7) and now state explicitly that each NN predicts multiple, related outputs, as detailed in Table 2. We chose a small set of task-specific networks rather than a single LSTM that ingests all inputs to predict all outputs because such monolithic models are prone to shortcut learning, which can induce implausible input–output relationships. By separating networks according to tasks with theoretically justified input–output connections, we introduce a soft causal regularization of the machine-learning component based on prior knowledge. This rationale is reflected in the manuscript (current preprint, lines 163–171).

**1.2.2**

You name your target variables for the ML tasks model constraints – but indeed there are no constraints in the model. Your constraints are predicted target variables by a ML algorithm and controlled by the performance of the ML model. What about out-of-sample inputs? They are not constrained and depend on the generality of your model.

Thank you for this point. In the revised manuscript, we adopt the term "data constraints" to better reflect their role in guiding predictions. Their influence indeed depends on the ML model's performance and generalization.

Regarding out-of-sample inputs, we assessed this with a new experiment that held out not only spatial blocks but also two full years (see our response to Section 1.2.13 for details). This comparison indicates robust generalisability, at least over short-term horizons (two years).

**1.2.3**

The Greek variables (e.g., alpha\_T) were trained by NNs – but it is not clear how you trained these parameters. Which target variable was used? These parameters seem to be hidden variables in the NNs, no target variables. Please be more precise about your ML architecture and a detailed ML model description. How is alpha\_T integrated in your NN?

Thank you for this question. We have added the following paragraph when we introduce the equation 1 to clarify this mechanism:

"Parameters such as  $\alpha_T$  are explicit NN outputs. During training, the NN predicts  $\alpha_T$  from its inputs, and this value—together with variables like fAPAR—is passed to the process-based component to estimate transpiration, which contributes to total ET. Because ET is constrained by observations, training minimizes the loss between predicted and observed ET, thereby adjusting  $\alpha_T$  iteratively. Moreover,  $\alpha_T$  is also shaped indirectly through other observational constraints (e.g., TWS, GPP), so multiple datasets jointly influence its learning."

For details on the full training procedure, please see our response in Section 1.2.7.

**1.2.4**

Please also clarify the CO2 dependency using beta in Eq. (2) so readers can understand how fertilization enters the model.

Thank you for this suggestion. We added the following to the revised manuscript when we introduce equation 2:

"... The  $\beta_{CO_2}$  term is a globally constant, trainable parameter that regulates the strength of the  $CO_2$  fertilization effect on GPP."

**1.2.5**

Also, how was the WUE learned in the model? On which spatial and temporal resolution are these parameters learned? I feel like having not enough information to fully understand your underlying ML architecture.

We thank the reviewer for this question. WUE is learned in the same way as  $\alpha_T$  (please see our responses in Section 1.2.3 and Section 1.2.7): the NN predicts WUE from inputs, which is then used in the process-based module and iteratively adjusted through observational constraints. WUE is estimated at daily temporal resolution and 1° spatial resolution. More generally, in Eqs. (1–4), variables with superscript < s > vary spatially but are temporally invariant, those with < s, t > vary across both space and time, and parameters without superscripts are globally constant. We highlight this in l.99 of the current version of the preprint.

**1.2.6**

145ff.: Are you using time series or only single time steps as input for your LSTM? I assume you were using time steps as the latter would not make sense. But it is not clearly described and misunderstanding in your description.

Thank you for pointing this out. H2CM uses full time series as input to the LSTMs, not isolated single steps. The LSTM processes the sequence recurrently, step by step. Our wording at line 146 ("at time step t...") was meant to illustrate the operation at one step, but the same

sequential processing applies across the entire series (t, t+1, t+2, ...). We revised the methods section (145ff) to clarify this and prevent misunderstanding by adding the following sentence:

"Note that Fig. 2 illustrates this process for a single time step; however, it is executed sequentially for each time step, starting from t = 0 until the final time step."

**1.2.7**

I understand that you used a simplified overview of your model architecture. But there is still missing more detailed information about the network architecture of you NN components. It would be helpful to have another figure especially for these components as well – as your model relies on the hybrid approach. How many layers, number neurons, training epochs, learning rate, any dropout or weight decay were used? How are the different NNs connected?

We thank the reviewer for this suggestion. We have prepared a new figure explaining the model architecture in much greater detail and included it in the Appendix for readers interested in these details:

**"Figure A1 illustrates the overall architecture of H2CM:**

- (A) Static inputs and compression. Panel A1 represents the static environmental inputs, including land cover, soil properties, wetland extent, and digital elevation. These features are passed through a fully connected neural network (FC-NN 1; A2) that compresses spatially varying but temporally invariant information into a latent vector. FC-NN 1 has two hidden layers with 150 and 12 units, respectively. The 12-unit latent vector is used as compressed static features and serves as a shared input to all dynamic sequence layers. The outputs of this static network (A3) include spatially varying parameters such as maximum soil moisture capacity  $(SM_{max})$  and  $\alpha_{Ei}$  controlling interception evaporation.
- (B–E) Dynamic sequence models. H2CM includes three long short-term memory (LSTM) networks and an additional fully connected neural network (FC-NN 2) to model time-varying processes. Each LSTM contains one hidden layer with 100 units and is connected to a small fully connected layer that transforms hidden states into physically interpretable parameters. These dynamic modules produce spatio-temporal predictions:
  - LSTM 1 (B1–B3): Receives the compressed static features together with dynamic drivers such as net radiation, precipitation, relative soil moisture, snow, groundwater, and fA-PAR at time t-1. It predicts coefficients ( $\alpha_{rsoil}$ ,  $\alpha_{rgw}$ ,  $\alpha_{smelt}$ ) that control soil recharge, groundwater recharge, and snowmelt processes.
  - LSTM 2 (C1–C3): Takes inputs including the static compressed vector, air temperature, vapor pressure deficit,  $CO_2$  concentration, relative soil moisture, and fAPAR and NPP at time t-1. It estimates carbon use efficiency and fAPAR.

- LSTM 3 (D1–D3): Uses static compressed features, net radiation, precipitation, fAPAR, and NPP at time t-1 to predict the basal respiration rate and  $\alpha_{Es}$  parameter that controls soil evaporation.
- FC-NN 2 (E1–E3): A fully connected neural network with two hidden layers (150 and 12 units) that predicts water use efficiency and the  $\alpha_T$  parameter, which represents effective conductance or stress response.
- (F) Global constants (learned). A set of globally learned parameters ( $Q_{10}$ ,  $\beta_{CO2}$ ,  $\beta_{\text{baseflow}}$ ,  $\beta_{snow}$ ) provide scaling relationships for temperature sensitivity of respiration,  $CO_2$  fertilization, baseflow recession, and correction of snowfall.
- (G) Coupled water—carbon cycle model. The outputs from the static and dynamic subnetworks are passed to a differentiable, process-based water—carbon cycle model that enforces mass balance between fluxes. This model represents the physical coupling between hydrological and carbon processes, ensuring consistency between water storage, evapotranspiration, carbon assimilation, and respiration.
- (H) Constrained spatio-temporal predictions. During training, the entire hybrid architecture—including the static and dynamic neural subnetworks and the differentiable process water—carbon cycle model—is optimized end-to-end. The neural network components generate spatially and spatio-temporally varying parameters that are passed into the process model, which produces simulated water fluxes and storages and carbon fluxes. Some of these outputs (net ecosystem exchange, gross primary productivity, fAPAR, terrestrial water storage, snow water equivalent, evapotranspiration, and runoff) are compared against observational targets through a composite loss function. Because the process-based component is fully differentiable, gradients of the loss propagate through the process equations back to the neural subnetworks via automatic differentiation. This enables the networks to learn physically consistent parameterizations that minimize discrepancies between modeled and observed dynamics."

Figure 1: Detailed overview of the H2CM framework. Static inputs (e.g., land cover, soil, elevation) are processed through a fully connected neural network (two hidden layers: 150 and 12 units) to generate compressed static features. These features feed into three LSTM networks (each with one hidden layer of 100 units) and an additional fully connected network that predict spatio-temporal parameters regulating hydrological and carbon processes, including  $\alpha_{rsoil}$ ,  $\alpha_{rgw}$ ,  $\alpha_{smelt}$ ,  $\alpha_{Es}$ ,  $\alpha_{Ei}$ , and  $\alpha_{T}$ . Globally learned constants  $(Q_{10}, \beta_{CO2}, \beta_{baseflow}, \beta_{snow})$  control temperature sensitivity,  $CO_2$  fertilization, baseflow recession, and snow corrections. Outputs from all subnetworks are coupled through a differentiable, mass-balanced water–carbon process model. The process layer produces water fluxes and storages and carbon fluxes, some of which are directly constrained by observations, including NEE, GPP, fAPAR, TWS, SWE, ET, and runoff.

Regarding hyperparameter tuning, such as the learning rate, we have added the following details to Section 2.3.3:

"... We set the initial learning rate to 0.005 and used a step-wise learning-rate scheduler (StepLR; (Paszke et al. 2019)) that decays the rate by a fixed factor at fixed epoch intervals. Hyperparameters were tuned by training multiple model variants and selecting the configuration that achieved strong validation performance with stable training. ..."

**You also mention a FCNN for data compression: What kind of architecture was used here? Was it an unsupervised approach?**

We thank the reviewer for this point. Briefly: the "compression" module is a fully connected network with two hidden layers (150 and 12 units) that maps static environmental inputs (land cover, soil properties, wetland extent, elevation) to a 12-dimensional latent vector. This vector serves as compressed static features and is shared across the dynamic sequence modules. It is not a separate unsupervised autoencoder; the compression is learned jointly with the rest of the model during training. For details, please see our answer in Section 1.2.7.

**1.2.9**

**I do not see any hyperparameter tuning in the manuscript. How were model hyperparameters chosen and/or validated?**

Please see the last paragraph of our answer in Section 1.2.7 where we have addressed this point.

**1.2.10**

In Tab. 3 WUE and CUE are defined as ratios – but in Tab. 2 these variables are defined as functions depending on multiple variables, trained by a NN. Please be more precise here on how the definitions are meant for you approach.

Thank you for raising this point. We agree that clarification is needed.

WUE and CUE are predicted directly by the NNs and then used in the process-based component to compute GPP and NPP, respectively. In Table 3, we show the conventional definitions:

- WUE = GPP / Transpiration
- CUE = NPP / GPP

In Table 2, these appear as NN outputs expressed as functions of specific input variables. To avoid confusion, in the revised manuscript we replaced WUE in Table 2 with  $\alpha_{WUE}$ .

Another difference is that Table 2 lists instantaneous, daily variables used as model inputs, whereas Table 3 reports annual means computed in post-processing from the model outputs.

The results are well presented. I am missing a short paragraph on the evaluation of the several trained NNs, for example on the performance of the WUE, CUE, etc. prediction alone. To increase confidence in the performance of H2CM, a brief description of the performance of the sub-variables would be helpful.

Thank you for highlighting the importance of evaluating the direct predictions of water—carbon cycle variables. Direct assessment of variables such as WUE and CUE is challenging due to the lack of large-scale observational datasets at the resolution of our model. These variables are usually derived indirectly through data-driven or process-based approaches. Therefore, in Section 3.1.4 (in the current version of the preprint) we chose to provide a qualitative evaluation by comparing our results with existing literature.

**1.2.12**

The NNs are trained by MSE Loss (Eq. 5) averaged equally over all data constraints. This implies that all constraints (TWS, SWE, ET, runoff, FAPAR, GPP, NEE, etc.) are treated with the same priority, regardless of their units or uncertainties. The authors should comment on this. might some constraints dominate the loss? Have the authors normalized each variable or adjusted for data uncertainty? Some acknowledgment of observational errors (and how they might affect the loss weighting) is appropriate.

Thank you for raising this point. To address unit differences, we apply a Z-transformation to both predicted and target variables before computing the loss (as discussed in lines 204–205 of the preprint). This centers the data at zero, scales it to unit variance, and removes unit dependence.

Currently, we do not apply explicit weighting across observational targets during optimization, so all constraints are treated with equal priority. This choice was based on the observation that performance across variables remained balanced, with the model capturing key patterns for each constraint.

Regarding observational uncertainty, we focus on robust patterns in the datasets. For example, FLUXCOM products are known to have uncertainty and bias in interannual variability of water and carbon fluxes. To reduce bias, we constrain the model against the mean seasonal cycle of GPP and ET from FLUXCOM-X-BASE rather than their full interannual variability (we discuss this in lines 88-89). While this approach mitigates bias, we acknowledge it does not fully resolve observational uncertainty. We already discuss this aspect in Section 3.4 of the preprint.

The 10-fold CV is spatial only. Thus, it is not clear how well the model would predict an unseen year (e.g., a future year). I encourage the authors to comment on this limitation. If possible, as a future step, holding out later years for test could provide insight into model stability under changing climate.

Thank you for raising this important point. We agree that it is currently unclear whether H2CM can generalize to unseen years. We chose spatial-only cross-validation due to inconsistencies in temporal coverage of observational datasets, which complicate a proper spatio-temporal split. For instance, one key carbon cycle constraint comes from OCO-2 inversions available only from 2014 onward, while our model uses data from 2001–2019. Holding out specific years would result in folds lacking critical data.

As an additional experiment to get a sense of whether the model can be applied to unseen years, we trained the model on years 2001–2017 and tested it on the subsequent 2 years (2018-2019) to assess generalization. We added the following text in the section where we describe our cross validation set up:

"... To assess the model's generalizability across both space and time, we conducted an additional experiment in which the model was trained using data from 2001 to 2017 and evaluated on the subsequent two years (2018–2019; Appendix D)."

In the section where we discuss results we added the following paragraph:

"Note that, although the results discussed here are based on the spatial-only cross-validation setup, Appendix D demonstrates that the model also generalizes well to unseen years, at least over short-term future periods."

And in Appendix, we added a new section:

"Figure D1 shows the performance comparison between spatially split and spatio-temporally split cross-validation folds using post-2017 time-series data. In the spatial split experiment, the model was trained on the complete 2001–2019 time series while holding out specific spatial grid cells for testing. In contrast, the spatio-temporal split experiment was trained on data from 2001–2017, with all data after 2017 withheld for spatio-temporal testing."

"Overall, the results suggest that the model maintains consistent performance when evaluated on the two unseen years, demonstrating generalisability not only across space but also over time—at least when tested on short-term future periods. In the spatio-temporal split experiment, neither the testing grid cells nor the final two years (2018–2019) were included in model training or validation, ensuring a fully unseen temporal and spatial domain during testing."

Figure 2: Model performance evaluated on the testing set comprising post-2017 time-series data, averaged over the testing grid cells. In the spatial split experiment, the model was trained on the full 2001–2019 dataset while holding out specific spatial grids for testing. In contrast, the spatio-temporal split experiment was trained only on data from 2001–2017, ensuring that no post-2017 information was included during training. Boxplots illustrate the distribution of errors across 10 cross-validation folds.

To you considered to use e.g., Physical Informed Neural Networks (PINNs) instead of simple FCNN to better control and constrain the physical processes behind?

Thank you for this suggestion. We did not use Physics-Informed Neural Networks (PINNs) in the current study, but we plan to explore their integration in future model developments.

Overall, the methodology is sound and described in good detail. Small clarifications and additional details (especially on the neural-network implementation) would improve reader understanding and reproducibility.

The authors treat global parameters e.g., beta as learnable. Section 3.1.5 shows the learned Q10 is about 1.24, which is lower than typical literature values (1.4–2). Similarly, the learned beeta values greatly exceed observational estimates. The authors rightly note this discrepancy and attribute it to equifinality and insufficient constraints. Please briefly discuss the implications: e.g., a high beta means the model might overestimate CO2 sensitivity if used for future scenarios. Emphasize that these global parameters are effectively unconstrained by data and could be fixed based on independent knowledge.

Thank you for this suggestion. To address this, we explored an alternative approach where strong priors—based on observational studies—are imposed on globally learnable parameters (see lines 321ff and Appendix C of the preprint). These experiments show that predictions closely follow the priors, indicating that equifinality prevents the model from departing significantly from these prescribed values.

We clarified this in the revised manuscript (Section 3.1.5):

"... These results indicate that, without such priors, the available data and process constraints leave the global parameters largely unconstrained; thus, knowledge-based priors are important, especially for analyses of future projections."

**1.3 Model evaluation and benchmarking**

**1.3.1**

Correlation and RMSE are mentioned, but it would help to provide bias or error values in the text or supplementary tables. E.g., "small RMSE for NEE IAV" (1.236), but exact numbers or global bias would be useful. A table summarizing global or zonal RMSE and bias for GPP, NEE, etc., in comparison to benchmarks would complement the discussion.

We thank the reviewer for this suggestion. We have added the following table in the Appendix, which presents global RMSE and bias values for carbon fluxes:

Table 1: Benchmarking metrics (RMSE and Bias) for modeled carbon fluxes against data constraints. Metrics are reported for gross primary productivity (GPP) and for net ecosystem exchange (NEE) benchmarked against OCO-2 and CarboScope inversion products. Each section (GPP, NEE (OCO-2), NEE (CarboScope)) summarizes results for Monthly, mean seasonal cycle (MSC), and Anomaly data. All metrics are based on globally averaged, area-weighted time series. Reported values represent the median across 10-fold cross-validation runs, with values in brackets indicating the minimum and maximum across folds.

| Metric              | Monthly             | MSC                 | Anomaly            |
|---------------------|---------------------|---------------------|--------------------|
| GPP                 |                     |                     |                    |
| RMSE [gC m-2 day-1] | 0.09, [0.07, 0.17]  | 0.08, [0.06, 0.16]  | 0.05, [0.04, 0.06] |
| Bias [gC m-2 day-1] | 0, [-0.15, 0.07]    | 0, [-0.15, 0.07]    | 0, [0, 0]          |
| NEE (OCO-2)         |                     |                     |                    |
| RMSE [gC m-2 day-1] | 0.04, [0.03, 0.06]  | 0.03, [0.02, 0.05]  | 0.02, [0.02, 0.02] |
| Bias [gC m-2 day-1] | 0, [-0.05, 0.03]    | 0, [-0.05, 0.03]    | 0, [0, 0]          |
| NEE (CarboScope)    |                     |                     |                    |
| RMSE [gC m-2 day-1] | 0.07, [0.06, 0.08]  | 0.06, [0.05, 0.07]  | 0.03, [0.03, 0.03] |
| Bias [gC m-2 day-1] | 0.02, [-0.02, 0.06] | 0.02, [-0.02, 0.06] | 0, [0, 0]          |

Additionally, we have included the following paragraph in Section 3.1.1 to explicitly report the relevant numbers:

"Globally, H2CM reproduces GPP patterns from FLUXCOM-X-BASE with RMSEs of 0.09, 0.08, and 0.05 gC  $m^{-2}$   $day^{-1}$  for monthly data, mean seasonal cycles (MSC), and monthly anomalies, respectively. For NEE from OCO-2 satellite inversions, RMSEs are 0.04, 0.03, and 0.02 gC  $m^{-2}$   $day^{-1}$  for the same categories. Compared to CarboScope in-situ inversions, H2CM yields RMSEs of 0.07, 0.06, and 0.03 gC  $m^{-2}$   $day^{-1}$ . Bias is negligible (close to 0) for GPP and NEE (OCO-2), and for NEE (CarboScope) is 0.02 gC  $m^{-2}$   $day^{-1}$  for monthly and MSC data, and zero for monthly anomalies (Table C1)."

**1.4 Reproducibility and transparency**

**1.4.1**

It would be helpful to have additional documentation (README, installation instructions) and example scripts/notebooks to run the model. Now, all daily outputs are shared.

Thank you for touching on this important aspect of reproducibility. We believe we have already addressed it by making all necessary components publicly available, including model code, inputs, observational targets, outputs, and documentation:

- Developmental model code (including a README.md with installation and run instructions): [GitHub: https://github.com/zavud/h2cm]
- Model version used in the preprint (including a README.md with installation and run instructions): [Zenodo, DOI: 10.5281/zenodo.15784689]
- Model inputs and observational targets: [Zenodo, DOI: 10.5281/zenodo.16575309]
- Daily model outputs: [Zenodo, DOI: 10.5281/zenodo.16572166]

**1.5 Interpretation and discussion of results**

**1.5.1**

The authors could strengthen the interpretation by commenting on potential future applications. E.g., since the model currently lacks an energy cycle (mentioned as future work), are there plans to incorporate dynamic vegetation or disturbances (aside from fire emissions)?

Thank you for raising this important point. We have broadened the last paragraph of the Conclusion section as follows:

"H2CM opens new avenues for studying the global carbon—water cycle and lays the ground-work for further development toward a hybrid land-surface model. Advancing in this direction will require integrating the surface energy cycle, incorporating dynamic vegetation and explicit carbon pools with turnover, representing additional key processes and disturbances (e.g., permafrost, fire, and land use/management), and increasing temporal resolution to resolve sub-daily dynamics. These are forward-looking requirements, and meeting them will be a substantial long-term effort beyond the scope of this study."

**1.6 Minor stuff**

**1.6.1**

**104: Please write Transpiration T to introduce the variable.**

Thank you for the suggestion. We explicitly introduced Transpiration (T).

**1.6.2**

Figure 4: Too small and the solid black background confuses. I suggest to make clean figures on white background. The title is also too small and does not fit to the explanations given in the caption. Please double check that your presented data fit to the presented titles in the figure.

Thank you for pointing this out. We revised Figure 4 accordingly.

**1.6.3**

In Tab. 1 the meteorological forcing data are described. Please briefly explain why you decided for this mixture of data sources.

Thank you for this suggestion. We agree that the rationale for selecting the meteorological forcing datasets should be clarified. We added a brief explanation in the Section 2.1:

"In general, we selected meteorological and static datasets that are widely used in the community, quality-checked, and offer the best compromise between spatial/temporal resolution and observational accuracy for each variable."

**1.6.4**

**100: You use the Greek letter for globally constant parameters. Does it include spatially and temporally constant?**

We thank the reviewer for pointing this out. Yes, globally constant learnable parameters (typically  $\beta$  parameters or  $Q_{10}$ ) are invariant across both space and time. Another way to recognise the variability in space and time is that globally constant parameters do not have superscript (such as s for space and s,t for space and time). We highlighted this aspect in the revised version (Section 2.2.1):

"... Parameters without any superscript represent globally constant parameters (denoted by the Greek letter  $\beta$  or Q10) that do not vary either in space or time and are learned by the neural network."

**1.6.5**

As the various datasets span different periods, the manuscript should explicitly state the time period used for training/evaluation. Ensure it is clear how these are aligned.

Thank you for this suggestion. We explicitly specified the temporal coverage of datasets in Table 1.

**1.6.6**

The authors may note that dynamic vegetation changes are not included due to static land use input, though FAPAR input does implicitly capture some phenological variability.

We thank the reviewer for this helpful suggestion. We have added a brief note where the equation for modeling transpiration is introduced:

"... Note that dynamic vegetation changes are not explicitly represented in H2CM; however, temporal variations in fAPAR provide an implicit representation of phenological changes."

**1.6.7**

Median and range of Q10 across folds is mentioned. It may be useful to similarly report the spread of prediction metrics across the 10 CV models. This would indicate robustness.

In the current version of the manuscript, we already include the Fig. 3 with cross bars showing the minimum, maximum, and average prediction metrics across the 10-fold cross-validation, based on spatially averaged values over the testing grid cells.

**1.6.8**

the conclusion asserts that H2CM "accurately reproduces the monthly patterns" and "global patterns" of GPP and NEE. While this is supported by the results, it may sound slightly overconfident given some know biases. Perhaps soften to "reproduces major features of the seasonal and spatial patterns…"

We agree with the reviewer and have revised the text to better reflect this point:

"... Our results indicate that the model captures the major features of the monthly, seasonal, interannual, and global (mean annual) patterns of both GPP and NEE."

**1.6.9**

Overall, the writing is professional and detailed, with only minor edits needed for polish.

Thank you for your positive feedback.

**1.7 Recommendation**

I recommend major revisions before acceptance based on the recommendations above. The reported revisions will strengthen the papers clarity and reproducibility but do not undermine the core findings.

Thank you for your thorough review and constructive feedback.

**2 Reviewer 2**

The manuscript "H2CM (v1.0): hybrid modelling of global water-carbon cycles..." by Baghirov et al. addresses a relevant and timely topic: the hybrid modelling of the land surface and terrestrial biosphere. It reports on the architecture, training and evaluation of a hybrid prototype. In principle, I consider the paper suitable for the journal.

I also have a substantial number of general and specific questions however that the current version leaves open. In my opinion, the manuscript would be much clearer and more useful if they are addressed in the general framing and writing.

Dear reviewer 2,

Thank you very much for taking the time to review our manuscript and for providing valuable feedback to help us improve its clarity.

**2.1 General points**

**2.1.1**

I find it hard to understand to what extent this model can actually be considered "hybrid". I hardly see any process-based components in the model description. There are equations 1-4, but they are highly simplistic and high-level multiplicative relationships, far simpler than the complexity of the machine learning components, or typical components of process-based land surface models.

We thank the reviewer for this insightful comment. We refer to H2CM as a hybrid model because it integrates machine learning with process-oriented principles. We agree that the process representations are conceptual rather than detailed parameterizations of sub-processes. This simplification reflects a deliberate trade-off between process complexity and parameter identifiability given the available data constraints, enabling the model to represent first-order processes while reducing uncertainty.

Importantly, the scope of H2CM is to provide an observation-constrained carbon—water reanalysis by fusing multiple data streams, rather than to serve as a fully mechanistic land-surface model for coupling within ESMs or for producing unconstrained long-horizon climate projections. Achieving those aims would require more complete and detailed process-based components. H2CM should therefore be regarded as a process-aware machine learning framework that links carbon and water data streams to provide a consistent, data-informed interpretation of their interactions, conceptually related to data-assimilation approaches.

In the revised manuscript, we added the following paragraph clarifying the rationale, design choices, and positioning of H2CM within the broader context of modeling and model—data integration strategies:

"H2CM is a process-aware hybrid framework that integrates machine learning with simplified process-based formulations, and can be positioned within the broader family of model-data integration strategies that aim to combine physics with machine learning. This family includes approaches such as physics-informed neural networks (Raissi, Perdikaris, and Karniadakis 2019; Tartakovsky et al. 2020; Wang et al. 2020), physics-guided machine learning (Khandelwal et al. 2020; Pawar et al. 2021; Karpatne et al. 2017), and differentiable modeling (Shen et al. 2023). Rather than reproducing the detailed parameterizations used in comprehensive landsurface models, H2CM employs conceptual process formulations to maintain interpretability and ensure parameter identifiability under the available observational constraints. The primary objective is to deliver a consistent, observation-constrained 'reanalysis' of coupled carbonwater states and fluxes over recent decades by fusing diverse data streams within a processinformed ML architecture. H2CM is not intended to replace or be coupled into Earth system models, nor to provide long-horizon projections; such applications would require more complete representations of processes and feedbacks. In this way, H2CM captures first-order processes while reducing uncertainty via data constraints, providing a bridge between empirical and process-based modeling and enabling a coherent interpretation of coupled carbon-water cycle variability and interactions."

**2.1.2**

Moreover, the model supposedly captures the "water cycle" and "carbon cycle". Besides the fact that cycles would include atmosphere and ocean (otherwise the cycle is not closed), the model does not seem to simulate any carbon pools only fluxes. If this model is supposed to be a step toward hybrid land surface modelling (that's how I understand the framing and motivation), what should be the approach to model differential equations where state variables have memory? How would one implement a similar model into an Earth system model, and what conclusions do the authors draw from their results to this end? What is it in the results that allows conclusions about the best approaches to such hybrid modelling?

We thank the reviewer very much for raising these points, which reflect a lack of clarity in the manuscript. As detailed in our response to the previous comment (Section 2.1.1), H2CM is intended for a "reanalysis" of the recent past rather than for integrating into Earth System Models for future projections. We apologize for this apparent misunderstanding, which is clarified by the section on the scope and rationale of H2CM (Section 2.1.1). This section includes a framing of H2CM within different emerging strategies of combining machine-learning with process-based modelling such as physics-informed neural networks.

Moreover, we expanded the discussion on the structural limitations of H2CM (the limitations section) from a process-representation perspective, emphasizing that explicit integration of carbon pools is essential for simulating longer-term dynamics, which are indeed a prerequisite for implementing a hybrid land model into an ESM:

"Currently, H2CM lacks several important components of the land system, which are relevant for the coupled water and carbon cycles. For example, it does not yet explicitly model vegetation and soil carbon pools, as the current focus has been on spatial variations of carbon fluxes from sub-seasonal to interannual time scales. Furthermore, H2CM does not incorporate the effects of disturbances such as fire, or other drivers including land-use change. These processes are critical for a comprehensive understanding of Earth system dynamics and would significantly enhance the applicability of our framework for studying broader system interactions. Addressing these limitations will be a key focus of future developments of H2CM."

Additionally, we added a paragraph to the conclusion section where we outline forward-looking requirements and challenges for evolving H2CM toward a hybrid land-surface model:

"H2CM opens new avenues for studying the global carbon—water cycle and lays the ground-work for further development toward a hybrid land-surface model. Advancing in this direction will require integrating the surface energy cycle, incorporating dynamic vegetation and explicit carbon pools with turnover, representing additional key processes and disturbances (e.g., permafrost, fire, and land use/management), and increasing temporal resolution to resolve sub-daily dynamics. These are forward-looking requirements, and meeting them will be a substantial long-term effort beyond the scope of this study."

While we acknowledge that the global carbon and water cycles encompass the ocean and atmosphere, it is common practice in the literature to use the term "terrestrial carbon—water cycle models" when referring specifically to the land component.

**2.1.3**

It is also unclear to me how soil moisture is modelled. There is reference to another recent study on what is called H2MV (Baghirov et al., 2025). I had a look there, but it seems to follow a similar approach in the sense that the model's mechanistic complexity and structure is rather simple, while model results seem to be mainly determined by the machine learning components.

We appreciate the reviewer's observation. Please also see our earlier responses to Section 2.1.1 and Section 2.1.2 regarding the overall scope and rationale of H2CM. Reviewer 2 is correct that the soil moisture module in H2CM adopts a relatively simple structural formulation. However, the machine learning component, in combination with observational constraints, can compensate for structural simplifications and enables the model to reproduce realistic and, in part, complex soil moisture dynamics. This behavior has been demonstrated and discussed in detail in our previous works, Kraft et al. (2021) and Baghirov et al. (2025).

**2.1.4**

Achieving a good match with observations with such a model is of course beneficial, but I wonder how well the model is able to extrapolate to different climates. For example, will it generate realistic trends when forced with data from the historical period over several decades, including the global warming trend? If not, why do we need a hybrid approach at all? To what extent do the process-based parts in the model contribute to the good performance? What makes H2CM better than Fluxcom-X-base in some cases – is it really the process-based part or is it a better machine learning approach or data? And whatever the answer is: Can the authors show this somehow? They say that a hybrid model is not a "black box" like ML models, so this may be possible? If the performance overall is largely determined by the data-driven parts (including the way different neural networks are combined), I wonder whether the framing of "hybrid modelling" is really helpful, in contrast to pure data-driven modelling with a specific architecture.

Reviewer 2 raises several relevant and insightful questions; however, many of these appear to stem from a misunderstanding regarding the scope and rationale of H2CM. These aspects are clarified in our responses to comments Section 2.1.1 and Section 2.1.2.

In the revised manuscript, we will further address the reviewer's specific points concerning the relative roles of the process-based, data-driven, and neural network components of the model. To do so, we will include a new paragraph that provides a qualitative comparison with other studies in the literature, as a comprehensive quantitative assessment would require extensive factorial experiments beyond the scope of this paper.

**2.1.5**

Regarding the general architecture of the model, Fig. 2 is helpful, but it is difficult for me to understand how the model is actually trained. The neural networks seem to generate inputs to what the authors call the "process-based water-carbon cycle model", which then generates observable variables. When the loss function is minimised during training, in what way is the process-based component used? Does it not need to backpropagate information somehow in order to feed back to

the neural networks and let them learn? Also, how do the authors use information on observational errors, specifically where different datasets on the same quantity (the two atmospheric co2 inversions) are used at the same time?

Please see our response to Reviewer 1 (Section 1.2.7), where we clarify the methodology by a new appendix section and accompanying figure to provide a more detailed explanation of the model architecture and training procedure.

Regarding the treatment of observational errors, we designed the training to emphasize the most robust aspects of each dataset rather than their full variability. For example, for FLUX-COM GPP we constrain the model only to the mean seasonal cycle, thereby avoiding potential biases from its uncertain interannual anomalies. For atmospheric  $CO_2$  inversions, we use the ensemble median of multiple OCO-2 inversion products to mitigate errors associated with individual inversion setups and transport models. In addition, we incorporate the globally averaged signal from the in situ-based CARBOSCOPE inversion, which provides a stable, large-scale constraint that is less sensitive to local retrieval uncertainties. The local, grid-scale patterns are primarily learned from the satellite-based inversion ensemble, which is available from 2014 onward following the launch of OCO-2. We acknowledge that residual observational uncertainties may still influence the model results, as discussed in Section 3.4 of the current preprint.

**2.1.6**

All observational datasets seem to always be used at the same time to train the model? Some parameters seem to be overconstrained. Which training data is actually important? How are physical constraints regarded, e.g. the conservation of mass? And why do the authors only train on a subset of grid cells but not time points?

Short answers is that yes, all available observational datasets are used simultaneously to constrain different aspects of the model. This multi-constraint setup is advantageous rather than problematic, as long as the constraints are valid — it helps the model become more robust and less sensitive to uncertainties or biases in any single dataset. Physical constraints are respected because the coupled water—carbon cycle model is fully differentiable and integrated end-to-end with the learning framework, allowing gradients to flow through all process-based components.

For further details, please see our responses to Reviewer 1 in Section 1.2.7 for details on how model training is implemented and how the process-based component is integrated, in Section 1.2.12 for the description of the loss function and the use of multiple observational datasets, and in Section 1.2.13 for the rationale behind spatial cross-validation and the new experiment where we held out several years to test model generalisability in future time.

**2.1.7**

Lastly, Section 3 in general shows seveal metrics, variables and regions, and evaluates H2CM. The choices of what to show here felt somewhat arbitrary to me, for example Sect. 3.3 and also Fig. 5. Why pick these examples? What is the key message that these results support? It would help if the authors presented clear arguments and criteria, and connected the results in an argumentative way.

We thank the reviewer for this question. Our goal in Section 3 was not only to test H2CM against its direct training data constraints—which we expect it to reproduce—but also to assess whether it generates plausible emergent patterns of water–carbon cycle dynamics that are not explicitly observed.

In Section 3.1.4 (corresponding to Figure 5) of the revised manuscript, we added the following paragraph to clarify the motivation for using global emerging patterns:

"In this section, we qualitatively evaluate emerging global patterns in H2CM, focusing on key indicators such as precipitation, water, carbon, and light-use efficiency. These metrics are essential for understanding water-carbon cycle dynamics and coupling, yet they are not directly observable at the global scale. Therefore, we draw on existing studies that have estimated these variables and qualitatively compare our results with their findings."

Regarding Section 3.3, we highlight a dry tropical region because such regions strongly influence global NEE interannual variability yet are often misrepresented in process-based models. Recent studies (Metz et al. 2023, 2025) have emphasized this issue. This analysis directly links to the previous section and the last panel of the figure: we now zoom into that region to infer mechanisms in H2CM, demonstrating that the model provides interpretability and potentially new insights. Accordingly, we added a brief paragraph in the revised version of the section to clarify this point.

"In this section we focus on a dry tropical region (Southern Africa) since these ecosystems exert strong control over global NEE interannual variability, yet process-based models often fail to capture their dynamics accurately (Metz et al. 2023, 2025)."

**2.2 More detailed points**

**2.2.1**

The authors say that H2CM is a "global" model. What does this mean? As far as I see, it is a local (grid cell specific) model without any spatial interactions, hence the domain and grid are arbitrary.

Thank you for raising this point. By "global" we mean that H2CM is trained on and applied to globally gridded observational datasets, with a single architecture used consistently across all land grid cells. We agree that the model is local and does not include explicit lateral

interactions. While, in principle, the domain and grid are arbitrary from a technical standpoint, the current design of the process equations, neural networks, and data streams is tailored to comparatively coarse grid cells consistent with the resolution and representativeness of the inputs, which avoids the need to model lateral interactions. We clarified this in the revised Introduction to prevent misunderstanding.

"H2CM is a global model in the sense that it is applied consistently across all terrestrial grid cells using global inputs and observations; it remains local and does not simulate lateral interactions among cells. Although the domain and grid are, in principle, arbitrary, the present design is tailored to comparatively coarse grid cells aligned with the input datasets, which mitigates the need to model lateral interactions."

**2.2.2**

Use of vocabulary: Note that the term grid refers to the spatial structuring of all grid cells. A grid cell refers to one spatial point. The authors often use "grid" even where they actually mean grid cell.

We appreciate the reviewer's observation. In the revised manuscript, we have carefully corrected the terminology, replacing "grid" with "grid cell(s)" where relevant to ensure consistent and accurate usage.

**2.2.3**

There are some typos; I suggest the authors read carefully before the next submission. Example: line 65-66: "the the", "objectives" (omit s), "withhold" (withheld), line 152 "compress"(es), line 263: "in in", Fig. B8 caption: "Runoff" should be lower case.

Thank you for pointing this out. All listed typos and other minor errors have been corrected in the revised manuscript.

**2.2.4**

Table 1: shortwave and longwave radiation seem to not be distinguished. But in practice, this will matter much for GPP and other fluxes. What is the underlying assumption here? Also, what is "short-term" versus "long-term" in the last two lines of the table? It could make sense to add a column showing the time period available for each dataset.

Thank you for this observation. Using net radiation as a single forcing is a conceptual simplification. We chose this because the NN is not sensitive to absolute levels and incoming shortwave and net radiation are empirically very tightly correlated at daily scales (median r =

0.96 across FLUXNET sites; Jung et al. (2024)), with differences likely within the uncertainty of radiation products. That said, the shortwave–longwave partitioning becomes important for long-term change (e.g., greenhouse-driven LW trends), and we plan to revise this when integrating the full surface energy cycle in the future. We clarified this assumption in the datasets section as follows:

"We use net radiation as a single forcing term rather than separately prescribing shortwave and longwave components. This simplification leverages the strong empirical correlation between shortwave and net radiation at daily scales (Jung et al. 2024)."

Regarding Table 1, we have revised it to explicitly show the temporal coverage for each dataset used.

**2.2.5**

**line 105 (Eq. 1): How is ETpot computed?**

Thank you for pointing this out. Potential evapotranspiration  $(ET_{pot})$  is computed based on available energy (net radiation) converted to a water flux by the latent heat of vaporization. The minimum of this value and the current soil moisture state is taken to prevent the possibility of negative soil moisture — a necessary condition that is only very rarely relevant. This detail was omitted here because the hydrological cycle is described in detail in a separate study (Baghirov et al. 2025); in this study, we focus on the carbon cycle and refer readers to Baghirov et al. (2025) for a full description of the water cycle components.

**2.2.6**

line 114-117, incl. Eq. 2: beta is supposed to capture the CO2 fertilisation effect, but it is just a constant, independent of CO2. The fertilisation effect is captured already by the linear dependence of GPP on CO2. What does this linear dependence imply when using the model for a transient situation with strongly increasing CO2? When considering all factors of Eq. 2, does the model generate a similar relationship as e.g. typical DGVMs?

We thank the reviewer for this thoughtful comment. The reviewer is correct that in Eq.(2),  $\beta_{CO_2}$  is constant and does not represent a dynamic  $CO_2$  fertilization effect. The explicit multiplication with atmospheric  $CO_2$  introduces a linear sensitivity. However, because WUE is learned by the neural network as a nonlinear function of meteorological drivers (soil moisture, VPD, radiation, etc.), emergent nonlinearities can arise when  $CO_2$  interacts with these factors. We chose a global constant for  $\beta_{CO_2}$  to reduce equifinality and improve parameter identifiability, as allowing both  $\beta_{CO_2}$  and WUE to vary freely in time could compromise interpretability.

In the revised manuscript, we included the following explanation when describing Eq. 2:

"... The  $\beta_{CO_2}$  term is a globally constant, trainable parameter that regulates the strength of the  $CO_2$  fertilization effect on GPP. Although  $\beta_{CO_2}$  does not explicitly represent a dynamic  $CO_2$  fertilization effect, the model's linear dependence on atmospheric  $CO_2$  interacts with  $\alpha_{WUE}$  which is learned by the neural network as a nonlinear function of relative soil moisture, vapor pressure deficit, net radiation, and static variables (Table 2)."

Regarding whether H2CM generates  $CO_2$  responses similar to typical DGVMs under strongly increasing  $CO_2$ , the patterns appear broadly comparable. For example, this can be seen in the global averages of monthly GPP anomalies over 2001–2019:

Figure 3: Globally, area-weighted averages of monthly GPP anomalies from H2CM and TRENDY. Lines indicate the median, while shaded areas represent the minimum and maximum across cross-validation folds for H2CM and ensemble members for TRENDY.

**2.2.7**

line 140: make clearer what you mean with labels "dynamic (recurrent)" and "static (fully connected)". Even though it may not be possible to draw the true architecture in Fig. 2, it would help to show different (idealised) icons for the NNs where these NNs have different architecture. If the figure becomes too busy: I don't think one actually needs to show global maps for all variables (which are

too small to see results anyway). This figure is about the structure not the actual data values.

We thank the reviewer for this suggestion and have revised the paragraph accordingly:

"H2CM consists of three primary modules: the dynamic (recurrent) module, which produces spatially and temporally varying variables; the static (fully connected) module, which generates spatially varying but temporally constant parameters; and the process-based (water-carbon cycle) module, which ensures mass conservation and governs the interactions between water and carbon fluxes:"

We have also revised the figure 2 to represent the dynamic module with an icon indicating recurrence, using a commonly recognized style.

**2.2.8**

line 184: I did not understand what the authors mean with "blocks". Are blocks the samples of 5x5 connected grid cells that are selected for training?

We have added a short explanatory sentence in the revised text to clarify the meaning of "blocks":

"... In this approach, "blocks" refer to spatially contiguous groups of  $5^{\circ} \times 5^{\circ}$  grid cells (25 grid cells in total) that are treated as single sampling units."

**2.2.9**

line 187-189: I is not really clear to me why validation on left-out time periods should not be possible.

Please see our response to Reviewer 1 in Section 1.2.13, where we have clarified this point.

**2.2.10**

line 191 and elsewhere. The authors cite Baghirov et al., 2025, but four references like that are listed in the reference list.

These citations are distinguished using lettered suffixes (2025, 2025a, 2025b, 2025c, 2025d) in both the text and reference list, following the journal's citation style. Each refers to a distinct resource (e.g., code release, data publication, or publication), and we therefore cite them separately as required.

line 196-197: Parameters theta and beta are adjusted – but how (see above)? How does training work involving the "process-based" model (whatever that is, also see above)?

Please see our responses to Reviewer 1 in Section 1.2.7 for details on model training.

**2.2.12**

line 199-201: What does it mean that the loss function is applied for each data constraint?! Isn't there one loss function where all different variables contribute? Or several loss terms? Then how to decide how important each loss is? Additionally, I don't understand why the Carboscope dataset is treated differently from all others.

We apologize for the confusion. We have revised the paragraph to clarify how the loss function is applied and to explain the specific treatment of the CarboScope dataset:

"The loss function combines all data constraints into a single objective, where MSE is computed across grid cells for each variable. The only exception is the long-term NEE interannual variance (CarboScope). In situ—based inversions such as CarboScope are generally more robust when averaged globally but become increasingly noisy at the grid-cell level. Therefore, we compute the global mean of both CarboScope and H2CM outputs during training within each batch and apply the loss to these global averages, ensuring the model fits large-scale interannual variability rather than local noise."

**2.2.13**

**line 204: perhaps briefly mention what a z-transformation is.**

We thank the reviewer for this suggestion. We have added a brief explanation of the Z-transformation in the revised text:

"Note that we apply a Z-transformation to predictions and observations before computing the loss. This standardizes each variable by subtracting its mean and dividing by its standard deviation, removing the effect of different units and balancing the contributions of each data constraint to the total loss."

**line 207: What is a "CV fold"?**

We added a brief clarification of "CV fold" at the first mention of the cross-validation setup, defining it as one of the subsets of data used for validation during the 10-fold cross-validation procedure:

"To evaluate the generalizability of H2CM, we use a 10-fold cross-validation (CV) setup. This process involves dividing the data into 10 subsets, or CV folds, and training 10 different models, each leaving out one fold as a validation set and using the remaining folds for training."

**2.2.15**

line 208-209: If all input is z-transformed, that means that all means are zero and standard deviation is 1? How then can the model be calibrated to respond to the correct mean values? For instance, how would the model respond to input temperature data that is 2°C higher than observed? This question also relates to the generalisability question above, and the question how the model responds to climate trends.

It is correct that z-transformation standardizes each input variable to have a mean of 0 and a standard deviation of 1, which facilitates training by placing all variables on a comparable scale. This transformation does not remove the physical meaning of the inputs. For example, if the temperature input is 2°C higher than the climatological mean, its z-transformed value reflects this deviation (in units of standard deviations), and the model responds accordingly. Regarding generalizability and climate trends, z-transformation does not prevent the model from capturing long-term trends, because deviations from the mean (e.g., warmer years or unusual events) are preserved in the standardized inputs. The learned relationships can therefore respond appropriately to both interannual variability and long-term trends.

We have briefly highlighted this point in the revised text:

"During training, all inputs to the neural networks are standardized using Z-transformation (so deviations from the mean are preserved in standardized units)."

**2.2.16**

line 223 (Eq. 7): This seems to be monthly anomalies. I would then not call that "Interannual variability"! And: If IAV is actually monthly variability, what is then the "monthly" values shown in Fig. 3? What is the difference? Is "monthly" the absolute data including seasonality, and "IAV" are the monthly anomalies?

We agree with the reviewer that the term monthly anomaly more accurately describes what is presented in Equation (7). The "monthly values" indeed represent the absolute data, which include the seasonal cycle, whereas the "monthly anomalies" refer to deviations from this seasonal mean. Accordingly, in the revised manuscript, we have replaced the term IAV with monthly anomalies throughout to ensure clarity and consistency.

**2.2.17**

3: (i) Why does the monthly data have much larger error than the monthly anomalies (IAV), whereas the other metrics look very good? (ii) Please make vertical axis ranges identical where possible. (iii) There is a lot of empty space in the figure, e.g. between bars. (iv) I don't understand the difference between the columns. The training data is always the same, and the authors evaluate different variables? Why then two columns for NEE? Does the training data differ? (v) What determines the range covered by the boxes? Maximum and minimum error from what distribution?

We thank the reviewer for these questions and suggestions:

(i) Larger RMSE values for the monthly data compared to the monthly anomalies arise because the monthly data include both the mean seasonal cycle and the anomalies. Consequently, mismatches in either the amplitude or the phase of the seasonal cycle between the model and observations substantially increase the overall RMSE. In contrast, the monthly anomalies represent data from which the dominant and predictable seasonal component has been removed, isolating only the year-to-year variations. This results in smaller RMSE values, as errors associated with seasonal mismatches are no longer present.

However, correlation metrics can show the opposite behavior. While the monthly data may exhibit high correlations due to the strong and well-aligned seasonal signal, the correlations for monthly anomalies are typically lower because they reflect the model's ability to capture the less regular, higher-frequency fluctuations around the mean seasonal cycle.

We have added the following text to the revised manuscript:

"In terms of RMSE, H2CM tends to exhibit higher errors for monthly data, followed by seasonal data, and then monthly anomaly. The higher RMSE in the monthly data reflects errors in reproducing both the amplitude and phase of the seasonal cycle. In contrast, the monthly anomalies exclude this seasonal component, resulting in smaller RMSE but typically lower correlations because only irregular year-to-year variations remain."

- (ii) Done.
- (iii) Done.
- (iv) Each column presents performance metrics for a different carbon-cycle data constraint (GPP or NEE), evaluated over the testing set (grid cells not used during training). The two

NEE columns correspond to distinct observational constraints used in H2CM: (1) short-term, spatially resolved NEE estimates from OCO-2, and (2) long-term, globally aggregated NEE monthly anomalies from CarboScope. These two datasets represent complementary observational constraints used for independent evaluation.

(v) The ranges shown by the cross bars represent the variability of model errors across the 10 cross-validation (CV) folds. Specifically, for each metric, we first compute the spatial average over the testing set in each CV fold, and then take the minimum and maximum of these fold-level mean errors. Thus, the cross bars span the range of model performance across folds, and the central lines indicate the mean across all folds. We have updated the figure caption.

**2.2.18**

4: (i) The grey colour makes it too hard to see the text. (ii) titles per panel or column would help. (iii) absolute GPP values are hard to compare between columns, perhaps add difference plots. (iv) What is meant by "members" in each case? Members from the 10 subsamples of grid cells when training H2CM? And in case of TRENDY are members the individual models? Does the map then show the median from all models at each grid cell, i.e. each grid cell comes from a different vegetation model?

We thank the reviewer for these suggestions:

- (i) Done.
- (ii) Done.
- (iii) We have added a difference map in the Appendix to illustrate spatial biases and have cross-referenced it in the main text where global patterns are discussed.
- (iv) H2CM: The "members" refer to the 10 subsamples of grid cells used in the 10-fold cross-validation. For each fold, a separate model is trained, producing predictions across the globe. The median across these 10 predictions is then taken for each grid cell. TRENDY: The "members" are the individual process-based models within the TRENDY ensemble. The median for each grid cell is calculated across these models. OCO-2: The "members" correspond to the different inversion ensemble members. Again, the median is calculated at each grid cell across members. We have improved the figure caption accordingly.

**2.2.19**

5: (i) too grey (see above). (ii) "emerging global patterns" in what data? The trained model I guess? (iii) What are "folds"? Is this figure meant to show how realistic H2CM output is? Then we would need to see observations as comparison. Or is this result meant to offer new insights into land-atmosphere physics? Then

this should be a clearer part of the framing in the abstract, introduction and conclusions.

We thank the reviewer for these suggestions:

- (i) We updated the color accordingly.
- (ii) We updated it to "Emerging global patterns in H2CM: ..."
- (iii) We have clarified this point (Section 2.2.14).

Please see our response to Section 2.1.7, which explains the rationale for presenting these patterns.

**2.2.20**

Sect 3.2: What is it that makes H2CM better than Fluxcom? Can the authors demonstrate this? What are the implications for hybrid land modelling in general?

Please see our detailed responses to Section 2.1.4 and Section 2.1.2.

**2.2.21**

**line 407: "the information is available" – which information?**

We thank the reviewer for raising this point and have revised the paragraph to clarify our intended meaning:

"H2CM can only fit the data constraints if two conditions are met: (1) the relevant signals are present in the meteorological forcing, meaning that the meteorological inputs (e.g., temperature, precipitation, radiation) contain the variability and information needed to inform the target carbon—water fluxes, and (2) the model's process formulations permit it. In other words, H2CM can only reproduce patterns in the data constraints if those patterns are encoded in the input drivers and can be captured by the model's process-based formulations. Consequently, H2CM does not perfectly fit the data constraints, even with its highly data-adaptive neural network component. This can, to some extent, limit the model's ability to adapt to uncertainties within the data constraints (Baghirov et al. 2025)."

line 408: "the model's process formulations permit it" – which process formulations? Is there evidence that they restrict the results in some way?

Please see our response to Section 2.2.21. Regarding the evidence, as shown in our results—particularly in the model's estimation of terrestrial water storage—H2CM cannot perfectly fit the observations (phase shift), despite its highly data-adaptive and complex neural network component. If we had used a purely machine-learning model with no process-based formulations, the neural network would fit the observations almost perfectly. This indicates that the model's process formulations do impose some restrictions on its predictions, at least to some extent.

**2.2.23**

line 411-414: How is the spread of results evidence for equifinality? Doesn't equifinality mean the opposite, i.e. that different parameters (different models) lead to the same result?

We agree that the spread among the 10 cross-validation models does not, by itself, demonstrate equifinality. Instead, it reflects prediction uncertainty arising from differences in training data, fold selection, and random weight initialization. We have updated the paragraph in the manuscript to clarify this and avoid potential misunderstanding:

"Lastly, our model is susceptible to the equifinality problem, where different processes or pathways can lead to similar outcomes, as is common in process-based models (Baghirov et al. 2025). While our 10-fold cross-validation does not directly measure equifinality, examining the spread among the 10 models trained on different data folds and with different random weight initializations provides insight into the uncertainty and robustness of the simulations under varying training data."

**2.2.24**

A1: (i) What is k1, k2...? Why "k"? Are these the samples used for training different versions of the model? Every block here is 5x5 grid cells? (ii) The testing set seems to be 1/11th of the data, i.e. not 10%. (iii) And what about the evaluation set mentioned in the text which should be another  $\sim 10\%$ ? (iv) What is a "fold"?

We thank the reviewer for these questions and suggestions:

(i) k1-k10 represent the 10 folds used in 10-fold cross-validation. Using "k" to denote folds is a very standard way in machine learning and statistics, where k refers to the number of folds. Each fold is a unique subset of the dataset. During cross-validation, each model is trained on

- 9 folds and validated on the remaining fold, resulting in 10 trained models. Yes, each block contains  $5\times5$  grid cells (please see our response to Section 2.2.8).
- (ii) We agree. So we revised the text in the cross validation strategy section slightly:
- "... These blocks are randomly selected from the global data, ensuring that the majority of the data (approximately 80% of the grid cells) is allocated to the training set. The remaining  $\sim 20\%$  of the data is reserved for validation and testing. In practice, each of the 10 cross-validation folds contains roughly 1/11 of the total data for validation, while  $\sim 1/11$  of the data is held out as a fixed testing set for final evaluation."
- (iii) Please see our response above (ii).
- (iv) Please see our response to Section 2.2.14.

B1-B8. (i) MSC is the time mean over the entire period? (ii) Do time series show a spatial average? Over what region? (iii) Why not the same period in all figures? Due to limited training data? (iv) Which of these time series are actually from the identical model and should be physically consistent? Maybe these can be put into one figure with several panels. (v) What is TWS in Fig. B5? Total water storage? Why does it differ so much from GRACE? Because of the low resolution of GRACE? (vi) In Fig. B6, SWE is snow water equivalent?

We thank the reviewer for these helpful questions and suggestions.

- (i) Yes, the MSC represents the time mean over the entire period of the respective data constraint. We have clarified this in the relevant figure captions.
- (ii) As indicated in the figure captions, the time series represent spatial averages over the testing set. For example, the caption in the preprint specifies: "Predicted versus target NEE (OCO-2) over the testing set (spatial domain) across 10 CV folds ...".
- (iii) Please see our responses to Section 1.6.5 and Section 2.2.4.
- (iv) All results are based on H2CM predictions and their corresponding observational datasets.
- (v) TWS refers to terrestrial water storage. We have clarified this in the figure caption. For discussion on differences between H2CM predictions and GRACE observations, please see our response to Section 2.2.22.
- (vi) Yes, SWE stands for snow water equivalent, and this has been clarified in the revised figure caption.

B9: again, remove white space to condense figure. "Water cycle constraints" – sounds like different constraints are used here compared to the other applications? Which data was used here? This should become clearer.

We thank the reviewer for pointing this out. We have condensed the figure by reducing white space and clarified the data constraint aspects in the revised manuscript.

**2.2.27**

Appendix C: Why is there a "prior" and a "posterior" parameter which sounds like Bayesian statistics language? The method described in the appendix rather seems to nudge a parameter toward a specific target value, instead of calibrating it after starting from an initial value.

We thank the reviewer for this observation. Indeed, our approach does not involve a full Bayesian inference framework. The term "prior" was used loosely to indicate external knowledge or plausible ranges for global constants, which are incorporated as constraints in the loss function. The penalty term acts as a regularization that nudges the estimated constants toward these target values, rather than performing a formal posterior update. We have revised the text in Appendix C to clarify this and avoid potential confusion with Bayesian terminology.

**2.2.28**

Some more info on the parameter calibration method would help.

Please see our response to Section 1.2.7.

**2.2.29**

line 466-468: The fact that the posterior equals the prior could also imply that the nudging (loss term) is just too strong? Why is it evidence for an underdetermined problem? Would it make sense to put a factor 0 < f < 1 in the definition of the loss term? Then the final parameters could be different?

We agree that the nudging term may currently be too strong, which could contribute to the posterior equaling the prior. However, we also interpret this behavior as indicative of underdetermination: if the observational constraints provided sufficient information to move the parameters away from the prior, the optimization would balance this trade-off despite the penalty. Introducing a scaling factor (0 < f < 1) for the nudging term, as you suggest, is a promising idea to test whether the current weighting is overly restrictive. We will consider this modification in future work, and we have clarified shortly these points in the revised manuscript:

"... We note that the observed equality between the estimations and reference values may partly reflect a strong nudging term in the loss function. At the same time, this behavior also reflects underdetermination: the observational data do not provide sufficient information to move the parameters substantially away from the prior. Without additional data or process constraints, H2CM's global constants remain underdetermined by the current observations and knowledge alone; therefore, including prior terms in the loss function is currently important."

**2.2.30**

**D1: What are transcom regions?**

Transcom regions refer to the geographic areas defined in the Transport and Climate Monitoring (TransCom) project, which is widely used in atmospheric science and climate studies (Baker et al. 2006). We clarified this in the figure caption.

**References**

- Baghirov, Zavud, Martin Jung, Markus Reichstein, Marco Körner, and Basil Kraft. 2025. "H2MV (V1.0): Global Physically Constrained Deep Learning Water Cycle Model with Vegetation." *Geoscientific Model Development* 18 (10): 2921–43. https://doi.org/10.5194/gmd-18-2921-2025.
- Baker, D. F., R. M. Law, K. R. Gurney, P. Rayner, P. Peylin, A. S. Denning, P. Bousquet, et al. 2006. "TransCom 3 Inversion Intercomparison: Impact of Transport Model Errors on the Interannual Variability of Regional CO2 Fluxes, 1988–2003." *Global Biogeochemical Cycles* 20 (1). https://doi.org/10.1029/2004gb002439.
- Jung, Martin, Sujan Koirala, Ulrich Weber, Kazuhito Ichii, Fabian Gans, Gustau Camps-Valls, Dario Papale, Christopher Schwalm, Gianluca Tramontana, and Markus Reichstein. 2019. "The FLUXCOM Ensemble of Global Land-Atmosphere Energy Fluxes." *Scientific Data* 6 (1): 74.
- Jung, Martin, Jacob Nelson, Mirco Migliavacca, Tarek El-Madany, Dario Papale, Markus Reichstein, Sophia Walther, and Thomas Wutzler. 2024. "Technical Note: Flagging Inconsistencies in Flux Tower Data." Biogeosciences 21 (7): 1827–46. https://doi.org/10.5194/bg-21-1827-2024.
- Jung, Martin, Christopher Schwalm, Mirco Migliavacca, Sophia Walther, Gustau Camps-Valls, Sujan Koirala, Peter Anthoni, et al. 2020. "Scaling Carbon Fluxes from Eddy Covariance Sites to Globe: Synthesis and Evaluation of the FLUXCOM Approach." Biogeosciences 17 (5): 1343–65.

- Karpatne, Anuj, Gowtham Atluri, James H. Faghmous, Michael Steinbach, Arindam Banerjee, Auroop Ganguly, Shashi Shekhar, Nagiza Samatova, and Vipin Kumar. 2017. "Theory-Guided Data Science: A New Paradigm for Scientific Discovery from Data." *IEEE Transactions on Knowledge and Data Engineering* 29 (10): 2318–31. https://doi.org/10.1109/TKDE.2017.2720168.
- Khandelwal, Ankush, Shaoming Xu, Xiang Li, Xiaowei Jia, Michael Stienbach, Christopher Duffy, John Nieber, and Vipin Kumar. 2020. "Physics Guided Machine Learning Methods for Hydrology." arXiv. https://doi.org/10.48550/ARXIV.2012.02854.
- Kraft, Basil, Martin Jung, Marco Körner, Sujan Koirala, and Markus Reichstein. 2021. "Towards Hybrid Modeling of the Global Hydrological Cycle." *Hydrology and Earth System Sciences Discussions* 2021: 1–40.
- Metz, Eva-Marie, Sanam N Vardag, Sourish Basu, Martin Jung, Bernhard Ahrens, Tarek El-Madany, Stephen Sitch, et al. 2023. "Soil Respiration—Driven CO2 Pulses Dominate Australia's Flux Variability." *Science* 379 (6639): 1332–35.
- Metz, Eva-Marie, Sanam Noreen Vardag, Sourish Basu, Martin Jung, and André Butz. 2025. "Seasonal and Interannual Variability in CO2 Fluxes in Southern Africa Seen by GOSAT." *Biogeosciences* 22 (2): 555–84.
- Nelson, Jacob A, Sophia Walther, Fabian Gans, Basil Kraft, Ulrich Weber, Kimberly Novick, Nina Buchmann, et al. 2024. "X-BASE: The First Terrestrial Carbon and Water Flux Products from an Extended Data-Driven Scaling Framework, FLUXCOM-x." Biogeosciences 21 (22): 5079–5115.
- Paszke, Adam, Sam Gross, Francisco Massa, Adam Lerer, James Bradbury, Gregory Chanan, Trevor Killeen, et al. 2019. "PyTorch: An Imperative Style, High-Performance Deep Learning Library." arXiv. https://doi.org/10.48550/ARXIV.1912.01703.
- Pawar, Suraj, Omer San, Burak Aksoylu, Adil Rasheed, and Trond Kvamsdal. 2021. "Physics Guided Machine Learning Using Simplified Theories." *Physics of Fluids* 33 (1). https://doi.org/10.1063/5.0038929.
- Raissi, M., P. Perdikaris, and G. E. Karniadakis. 2019. "Physics-Informed Neural Networks: A Deep Learning Framework for Solving Forward and Inverse Problems Involving Nonlinear Partial Differential Equations." *Journal of Computational Physics* 378 (February): 686–707. https://doi.org/10.1016/j.jcp.2018.10.045.
- Shen, Chaopeng, Alison P Appling, Pierre Gentine, Toshiyuki Bandai, Hoshin Gupta, Alexandre Tartakovsky, Marco Baity-Jesi, et al. 2023. "Differentiable Modelling to Unify Machine Learning and Physical Models for Geosciences." Nature Reviews Earth & Environment 4 (8): 552–67.
- Sitch, Stephen, Pierre Friedlingstein, Nicolas Gruber, Steve D Jones, Guillermo Murray-Tortarolo, Anders Ahlström, Scott C Doney, et al. 2015. "Recent Trends and Drivers of Regional Sources and Sinks of Carbon Dioxide." *Biogeosciences* 12 (3): 653–79.
- Sitch, Stephen, Michael O'sullivan, Eddy Robertson, Pierre Friedlingstein, Clément Albergel, Peter Anthoni, Almut Arneth, et al. 2024. "Trends and Drivers of Terrestrial Sources and Sinks of Carbon Dioxide: An Overview of the TRENDY Project." Global Biogeochemical Cycles 38 (7): e2024GB008102.
- Tartakovsky, A. M., C. Ortiz Marrero, Paris Perdikaris, G. D. Tartakovsky, and D. Barajas-Solano. 2020. "Physics-informed Deep Neural Networks for Learning Parameters and Constitutive Relationships in Subsurface Flow Problems." Water Resources Research 56 (5). https://doi.org/10.1029/2019wr026731.

Wang, Nanzhe, Dongxiao Zhang, Haibin Chang, and Heng Li. 2020. "Deep Learning of Subsurface Flow via Theory-Guided Neural Network." *Journal of Hydrology* 584 (May): 124700. https://doi.org/10.1016/j.jhydrol.2020.124700.